# A plasmid-encoded peptide from *Staphylococcus aureus* induces anti-myeloperoxidase nephritogenic autoimmunity

Joshua D. Ooi[1], Jhih-Hang Jiang [2], Peter J. Eggenhuizen [1], Ling L. Chua [1,14], Mirjan van Timmeren[3], Khai L. Loh [4], Kim M. O'Sullivan[1], Poh Y. Gan[1], Yong Zhong[1], Kirill Tsyganov [5], Lani R. Shochet[1,6], Jessica Ryan[1,6], Coen A. Stegeman[7], Lars Fugger [8], Hugh H. Reid [4,9], Jamie Rossjohn [4,9,10], Peter Heeringa [3], Stephen R. Holdsworth[1,6], Anton Y. Peleg [2,11] & A. Richard Kitching [1,6,12,13]

Autoreactivity to myeloperoxidase (MPO) causes anti-neutrophil cytoplasmic antibody (ANCA)-associated vasculitis (AAV), with rapidly progressive glomerulonephritis. Here, we show that a *Staphylococcus aureus* peptide, homologous to an immunodominant MPO T-cell epitope (MPO$_{409-428}$), can induce anti-MPO autoimmunity. The peptide (6PGD$_{391-410}$) is part of a plasmid-encoded 6-phosphogluconate dehydrogenase found in some *S. aureus* strains. It induces anti-MPO T-cell autoimmunity and MPO-ANCA in mice, whereas related sequences do not. Mice immunized with 6PGD$_{391-410}$, or with *S. aureus* containing a plasmid expressing 6PGD$_{391-410}$, develop glomerulonephritis when MPO is deposited in glomeruli. The peptide induces anti-MPO autoreactivity in the context of three MHC class II allomorphs. Furthermore, we show that 6PGD$_{391-410}$ is immunogenic in humans, as healthy human and AAV patient sera contain anti-6PGD and anti-6PGD$_{391-410}$ antibodies. Therefore, our results support the idea that bacterial plasmids might have a function in autoimmune disease.

[1] Centre for Inflammatory Diseases, Monash University Department of Medicine, Monash Medical Centre, Clayton, VIC 3168, Australia. [2] Infection and Immunity Program, Monash Biomedicine Discovery Institute and Department of Microbiology, Monash University, Clayton, VIC 3800, Australia. [3] Department of Pathology and Medical Biology, University of Groningen, University Medical Center Groningen, Groningen 9700 RB, The Netherlands. [4] Infection and Immunity Program and Department of Biochemistry and Molecular Biology, Biomedicine Discovery Institute, Monash University, Clayton, VIC 3800, Australia. [5] Monash Bioinformatics Platform, Monash University, Clayton, VIC 3800, Australia. [6] Department of Nephrology, Monash Health, Clayton, VIC 3168, Australia. [7] Department of Internal Medicine, Division of Nephrology, University of Groningen, University Medical Center Groningen, Groningen 9700 RB, The Netherlands. [8] Oxford Centre for Neuroinflammation, Nuffield Department of Clinical Neurosciences, and MRC Human Immunology Unit, Weatherall Institute of Molecular Medicine, John Radcliffe Hospital, University of Oxford, Oxford OX3 9DS, UK. [9] Australian Research Council Centre of Excellence in Advanced Molecular Imaging, Monash University, Clayton, VIC 3800, Australia. [10] Institute of Infection and Immunity, School of Medicine, Cardiff University, Cardiff CF14-4XN, UK. [11] Department of Infectious Diseases, Alfred Hospital and Central Clinical School, Monash University, Melbourne, VIC 3004, Australia. [12] NHMRC Centre for Personalised Immunology, Monash University, Clayton, VIC 3168, Australia. [13] Department of Pediatric Nephrology, Monash Health, Clayton, VIC 3168, Australia. [14] Present address: Department of Paediatrics, Faculty of Medicine, University of Malaya, Kuala Lumpur 50603, Malaysia. Correspondence and requests for materials should be addressed to A.R.K. (email: richard.kitching@monash.edu)

Loss of tolerance to the neutrophil enzyme myeloperoxidase (MPO) leads to anti-neutrophil cytoplasmic antibody (ANCA)-associated vasculitis (MPO-AAV), an autoimmune disease that can affect multiple tissues but which often involves the kidney. In MPO-AAV, patients frequently develop rapidly progressive glomerulonephritis and are at risk of end-stage kidney failure[1]. The other major autoantigen known to be clinically relevant in AAV is the neutrophil serine protease, proteinase-3 (PR3). MPO-AAV and PR3-AAV, while having some differences, share similar pathogenic features. In MPO-AAV, tissue injury is induced not by autoantibodies binding to target tissues such as the kidney, but by anti-MPO autoantibodies (MPO-ANCA) that bind to and activate neutrophils causing glomerular neutrophil recruitment, degranulation, and NETosis[2–4]. These activated neutrophils are not only themselves responsible for significant tissue injury and damage, they also deposit MPO in and around glomerular capillaries[2,4–6]. Thus, MPO accumulating in glomeruli may function as an antigenic target for MPO-specific effector CD4[+] and CD8[+] T cells that induce a further wave of cell-mediated injury[3,6–9].

Although it is unclear how tolerance to neutrophil cytoplasmic antigens MPO and proteinase-3 (PR3) is lost and how disease is triggered[10], like many autoimmune diseases[11], both genetic and environmental factors are probably important[12,13]. In particular, infection has been implicated both in clinical studies, and in in vitro and in vivo experimental work[5,14–17]. Nasal carriage of Staphylococcus aureus is associated with an increase in relapse of disease in granulomatosis with polyangiitis, characterized by loss of tolerance to PR3 (PR3-AAV)[14]. Less is known about S. aureus colonization of people with MPO-AAV. While chronic nasal carriage is uncommon in those with microscopic polyangiitis and renal limited vasculitis, usually associated with MPO-ANCA[18], nasal colonization does occur[19] and case reports implicate S. aureus in the development of this condition[20–22]. There are several mechanisms by which infections might influence AAV; superantigens have been hypothesized to have a function[23], and pathogen-associated molecular patterns stimulate antigen presentation, B cells, and prime neutrophils[24]. The release of auto-antigens (including PR3 and MPO) by neutrophils at sites of infection might also affect the maintenance of tolerance. A further potential consequence of the uptake of neutrophil-derived auto-antigens by antigen-presenting cells at sites of inflammation with innate immune system activation could be the development of molecular mimicry. As molecular mimicry can lead to T-cell receptor (TCR) cross-reactivity[25–28], a microbial mimotope presented as a peptide by MHC Class II (MHCII) might activate TCRs that also recognize PR3 or MPO-derived epitopes presented by MHCII.

Some evidence supports the involvement of molecular mimicry in the loss of tolerance to neutrophil antigens in AAV. The complementary PR3 autoantigenic sequence, implicated in loss of tolerance to PR3, shares homology with bacterial peptides, including some from S. aureus[29]. Another target neutrophil antigen, lysosomal antigen membrane protein-2 (LAMP-2), shares sequence homology with the bacterial adhesin FimH, with FimH immunization of rats inducing anti-LAMP-2 auto-antibodies and glomerulonephritis[30]. However, it is not known whether molecular mimicry has any function in loss of tolerance to MPO and the resultant development of MPO-AAV.

Here, we demonstrate that molecular mimicry mechanistically contributes to the loss of tolerance to MPO in AAV. We evaluate whether microbial-derived peptides, including those from S. aureus, with sequence homology to the immunodominant MPO CD4[+] T-cell epitope can induce the expansion of naive CD4[+] T cells that recognize MPO, with the subsequent development of cross-reactive anti-MPO autoimmunity leading to glomerulonephritis and AAV. We identify a S. aureus peptide, 6-phosphogluconate dehydrogenase (6PGD)$_{391–410}$ derived from a plasmid-encoded protein that induces cellular and humoral anti-MPO autoimmunity and experimental anti-MPO glomerulonephritis. Thus, molecular mimicry mediated by a bacterial plasmid capable of horizontal transmission represents a potential mechanism of loss of tolerance in autoimmune disease.

## Results

**Highly homologous peptides do not induce autoreactivity**. To determine if autoreactivity to the immunodominant MPO CD4[+] T-cell epitope, mouse MPO$_{409–428}$[6], could be induced by microbial peptides, we performed a protein BLAST (blastp) search using the core 11-mer sequence of the equivalent human MPO peptide, $_{441}$RLYQEARKIVG$_{451}$ (mouse MPO peptide sequence and numbering: $_{415}$KLYQEARKIVG$_{425}$). Sequences from the Animalia kingdom (taxid:33208) and microbes not known to colonize humans were excluded. Based on the search results, we selected the four most homologous sequences (Supplementary Table 1) and because we have demonstrated previously that a 20-mer peptide induces stronger immunoreactivity to MPO[6] (concordant with MHCII molecules having an open ended binding groove[31]), we synthesized 20-mers based on the four identified sequences. For example, for the Aspergillus fumigatus HEAT repeat protein$_{831–841}$ ($_{831}$RWYQEARKIIF$_{841}$) the synthesized 20-mer was $_{825}$ISALPQRWYQEARKIIFEAA$_{844}$. To determine whether these sequences could induce anti-MPO autoimmunity we immunized C57BL/6 mice with individual 20-mers and measured T-cell reactivity to either the immunizing peptide, MPO$_{409–428}$, or recombinant mouse (rm)MPO using interferon-γ (IFN-γ) and interleukin (IL)-17A ELISPOTs and [3H]-T proliferation assays. While some homologous sequences induced reactivity to themselves, none induced reactivity to MPO$_{409–428}$ or whole rmMPO (Fig. 1a–f), demonstrating that high sequence homology per se does not result in immunological cross-reactivity to MPO.

**A S. aureus-derived peptide induces anti-MPO autoimmunity**. As S. aureus infections can precede the development of MPO-AAV[20–22], they are related to an overlapping form of vasculitis (PR3-AAV)[14,29] and nasal colonization of S. aureus has been found in people with MPO-AAV[19] we identified a S. aureus-derived peptide with sequence homology to human MPO$_{441–451}$ by protein BLAST. The highest scoring S. aureus-derived peptide containing the previously defined critical MPO$_{441–451}$ T-cell epitope residues (Tyr443, Arg447, Ile449 and Val450: $_{441}$RLYQEARKIVG$_{451}$)[6] was selected (BLAST MAX score of 18.0 out of 38.4 compared to human MPO$_{441–451}$). The identified peptide, 6PGD$_{397–408}$ ($_{397}$TDYQEALRDVVA$_{408}$) was from 6-phosphogluconate dehydrogenase (6PGD), an enzyme of the pentose phosphate pathway, and was first described within the plasmid pSJH101 from the clinically relevant S. aureus strain JH1[32]. To determine whether this 6PGD$_{397–408}$ sequence induced autoimmunity to MPO, we immunized C57BL/6 mice with 6PGD$_{391–410}$ ($_{391}$YFKNIVTDYQEALRDVVATG$_{410}$). Mice developed reactivity to 6PGD$_{391–410}$, as well as autoreactivity to both the immunodominant MPO CD4[+] T-cell epitope, MPO$_{409–428}$, and to rmMPO (Fig. 2a). MPO$_{409–428}$-immunized mice served as a positive control. To determine if exposure to 6PGD$_{391–410}$ induces in vivo expansion of MPO-specific T cells, we immunized mice with 6PGD$_{391–410}$ then enumerated the number of MPO-specific T cells using an I-A[b] tetramer presenting the core mouse MPO T-cell epitope ($_{415}$KLYQEARKIVG$_{425}$). We compared the total numbers of MPO-specific T cells from naive mice, OVA$_{323–339}$ immunized mice and

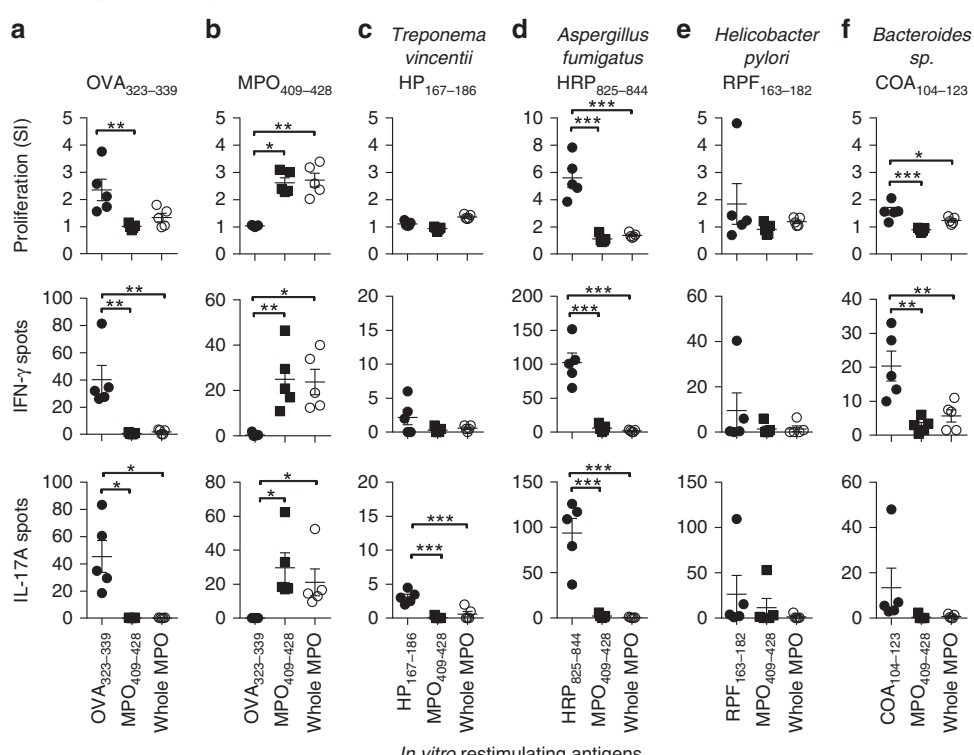

**Fig. 1** Microbe-derived peptides with closest sequence homology to MPO409–428 do not induce cross-reactivity to MPO. C57BL/6 mice (n = 5 each group) were immunized with peptides, either **a** OVA323–339 (negative control), **b** MPO409–428 (positive control), **c** *Treponema vincentii*-derived hypothetical protein, HP167–186, **d** *Aspergillus fumigatus*-derived HEAT repeat protein, HRP825–844, **e** *Helicobacter pylori*-derived RNA polymerase factor sigma-54, RPF163–182, or **f** *Bacteroides sp.*-derived chloramphenicol O-acetyltransferase, COA104–123, then, T-cell recall responses measured ex vivo by restimulating draining lymph node cells with either the immunizing peptide, MPO409–428 or recombinant mouse MPO (rmMPO) by [3H]-thymidine proliferation assays (top row), and ELISPOT for IFN-γ (middle row) or IL-17A (bottom row). Each dot represents the response from one mouse, error bars are the mean ± s.e.m. Data are representative of two independent experiments. *P < 0.05, **P < 0.01, ***P < 0.001 by Kruskal–Wallis test. Source data are provided as a Source Data file

MPO409–428 immunized mice using MPO:I-A^b tetramers. Cells were tetramer enriched using magnetic beads, then gated on live, CD4+, Dump−, MPO:I-A^b tetramer+ cells. Compared with naive mice and with mice immunized with OVA323–339, mice immunized with 6PGD391–410 exhibited a ~ 30-fold increase in MPO:I-A^b-specific CD4+ T cells (Fig. 2b). Thus, 6PGD391–410 induces expansion of MPO415–425-specific CD4+ cells and pro-inflammatory autoreactivity to MPO.

Serum from 6PGD391–410 immunized mice bound to fixed thioglycolate induced peritoneal neutrophils from C57BL/6 mice, in a perinuclear ANCA (pANCA) fashion (Fig. 3a) but not to MPO deficient (Mpo−/−) mouse neutrophils, and to whole native mouse (nm)MPO by enzyme-linked immune sorbent assay (ELISA) (Fig. 3b), findings that meet the diagnostic criteria for MPO-ANCA positivity in humans[33]. Furthermore, purified serum IgG bound to the clinically relevant human linear B-cell epitope MPO447–459 (Fig. 3c)[34]. To demonstrate antibody cross-reactivity between 6PGD391–410 and MPO409–428, we performed an inhibition ELISA. Purified serum IgG from 6PGD391–410 immunized mice was pre-incubated with 6PGD391–410, then used to detect anti-MPO409–428 IgG by ELISA. Serum IgG from *S. aureus* 6PGD391–410 immunized mice pre-incubated with *S. aureus* 6PGD391–410 had lower antibody titers compared with serum IgG pre-incubated with blocking buffer only (Fig. 3d). These cross-reacting antibodies were functionally active, as serum IgG from 6PGD391–410 immunized mice induced reactive oxygen species production from LPS-primed bone marrow mouse neutrophils in vitro as detected by the conversion of

dihydrorhodamine to rhodamine 123 (Fig. 3e). In vivo, passive transfer of this IgG fraction induced acute neutrophil glomerular recruitment in LPS-primed C57BL/6 mice, albeit at a low level (Fig. 3f). These data demonstrate that antibodies specific for *S. aureus* 6PGD391–410 cross-react with MPO409–428 and that the *S. aureus*-derived peptide induces both anti-MPO T-cell autoreactivity and biologically active MPO-ANCA.

To identify if the *S. aureus*-derived 6PGD protein is immunoreactive in healthy humans and in AAV patients, we measured IgG antibodies specific for the *S. aureus* pSJH101 6PGD protein by ELISA in sera from a Groningen cohort of healthy human subjects, 31 MPO-AAV patients and 30 PR3-AAV patients. We found detectable levels of *S. aureus* 6PGD-specific IgG in all three groups (Fig. 4a) implying that *S. aureus* pSJH101 6PGD is an immunogenic protein in humans. Furthermore, sera exhibited reactivity to the pSJH101 JH1 *S. aureus* 6PGD391–410 sequence by ELISA (Fig. 4b), demonstrating the immunogenicity of this sequence in humans. There were no significant differences in antibody titers between groups. To identify whether 6PGD391–410 can cross-react with anti-MPO antibodies in acute MPO-AAV, a Monash cohort of 15 patients with acute, active MPO-AAV was assessed (Supplementary Table 2). Purified IgG from these patients was assessed by inhibition ELISA by pre-incubation with 6PGD391–410, then antibodies to human MPO435–454 (the homologous sequence to mouse MPO409–428) were examined by ELISA. Of the 15 patients, five showed a significant reduction in anti-human MPO435–454 titers after incubation with 6PGD391–410 (Fig. 4c).

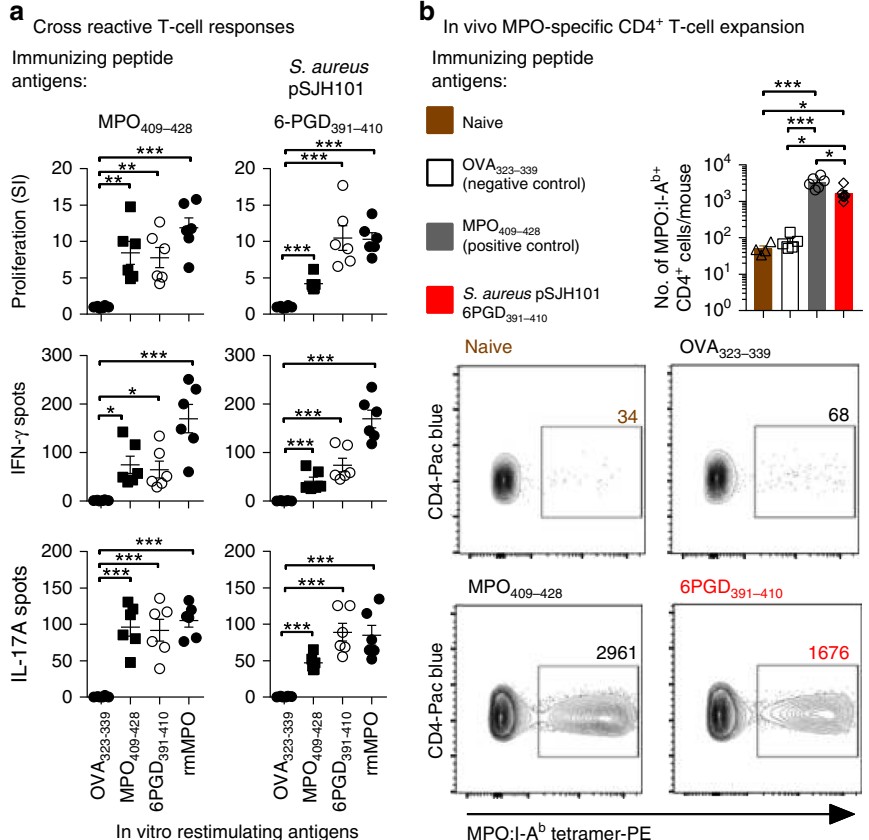

**Fig. 2** Immunization with *S. aureus* pSJH101-derived 6PGD$_{391-410}$ induces anti-MPO T-cell responses. **a** C57BL/6 mice ($n = 6$ each group) were immunized with either MPO$_{409-428}$ or *S. aureus* pSJH101-derived 6PGD$_{391-410}$, then T-cell responses measured ex vivo to either OVA$_{323-339}$, MPO$_{409-428}$, 6PGD$_{391-410}$, or recombinant mouse MPO (rmMPO) using [$^3$H]-thymidine proliferation assays (top row), and ELISPOT for IFN-γ (middle row) or IL-17A (bottom row). Each dot represents one mouse; data are representative of two independent experiments. **b** In vivo expansion of MPO-specific CD4$^+$ T cells. Cells from lymph nodes and spleen of C57BL/6 mice, either naive ($n = 4$), immunized with OVA$_{323-339}$ ($n = 5$), MPO$_{409-428}$ ($n = 6$) or *S. aureus* pSJH101-derived 6PGD$_{391-410}$ ($n = 6$). Results are expressed as number of MPO:I-A$^b$ tetramer$^+$ cells per mouse. Error bars represent the mean ± s.e.m. *$P < 0.05$, **$P < 0.01$, ***$P < 0.001$ by Kruskal–Wallis test. Source data are provided as a Source Data file

**S. aureus clonal specificity for the 6PGD$_{397-408}$ mimotope**. This particular 6PGD$_{397-408}$ sequence is unique to the *Staphylococcus* genus. *S. aureus* makes up the majority of publicly available staphylococcal genomes and the 6PGD$_{397-408}$ sequence of interest predominates in a clinically relevant *S. aureus* clonal complex (CC) known as CC5, to which the JH-1 strain belongs[35,36]. We assessed the multi-locus sequence type of 136 of the 143 sequenced *S. aureus* strains containing the 6PGD$_{397-408}$ mimic sequence and found that 115 (85% of those typed) of them were CC5. *S. aureus* CC5 strains have been described in Asia, America, Australia, Africa, and Europe[35–37]. There are 2544 publicly available CC5 *S. aureus* genomes, indicating that ~ 5% of sequenced CC5 strains contain the 6PGD$_{397-408}$ sequence. To assess the specificity of the CC5 related 6PGD$_{397-408}$ sequence in inducing cross-reactivity to MPO, we selected the four 6PGD variants most homologous to the pSJH101-derived *S. aureus* sequence (Supplementary Table 3), commonly found in sequenced *S. aureus* genomes. While each 6PGD peptide induced T-cell reactivity to itself, remarkably, none induced cross-reactive anti-MPO T-cell responses (Fig. 5a–f). Mice immunized with these variants did not develop MPO-ANCA, either by indirect immunofluorescence on mouse neutrophils (Fig. 6a) or by ELISA (Fig. 6b). When we measured anti-MPO$_{447-459}$-specific IgG in purified serum IgG, detectable levels of IgG were found only in Variant 1, but not in any of the other 6PGD peptide variants (Fig. 6c). Therefore, while the variant sequences of this *S. aureus*

6PGD-derived peptide are immunogenic, it is only the JH1, pSJH101 6PGD$_{397-408}$ sequence that induces anti-MPO T-cell responses and MPO-ANCA. To exclude the possibility that the orthologous, but dissimilar mammalian 6PGD sequence (6PGD$_{394-413}$) itself represented a new autoimmune target, mice were immunized with mouse 6PGD$_{394-413}$. This sequence did not induce cross-reactivity to MPO$_{409-428}$ or whole MPO (Supplementary Fig. 1).

**Immunization with 6PGD$_{391-410}$ leads to anti-MPO nephritis**. To determine if the loss of tolerance to MPO induced by *S. aureus* JH1-derived pSJH101 6PGD$_{391-410}$ could result in anti-MPO glomerulonephritis, we used our established model of T-cell-mediated anti-MPO glomerulonephritis[9,38]. In this model, C57BL/6 mice immunized with MPO lose tolerance to MPO but do not develop ANCA of sufficient pathogenicity to induce glomerulonephritis. Therefore, MPO is deposited within the glomerulus via neutrophils transiently recruited by injection of low dose of heterologous anti-mouse basement membrane globulin. In this context, effector MPO-specific T cells recognize MPO peptides and mediate glomerular injury[6,8,9]. MPO-immunized mice develop glomerulonephritis with pathological albuminuria and segmental glomerular necrosis. Using this protocol, mice immunized with the *S. aureus* JH1-derived pSJH101 6PGD$_{391-410}$ peptide developed glomerulonephritis of similar severity to MPO-

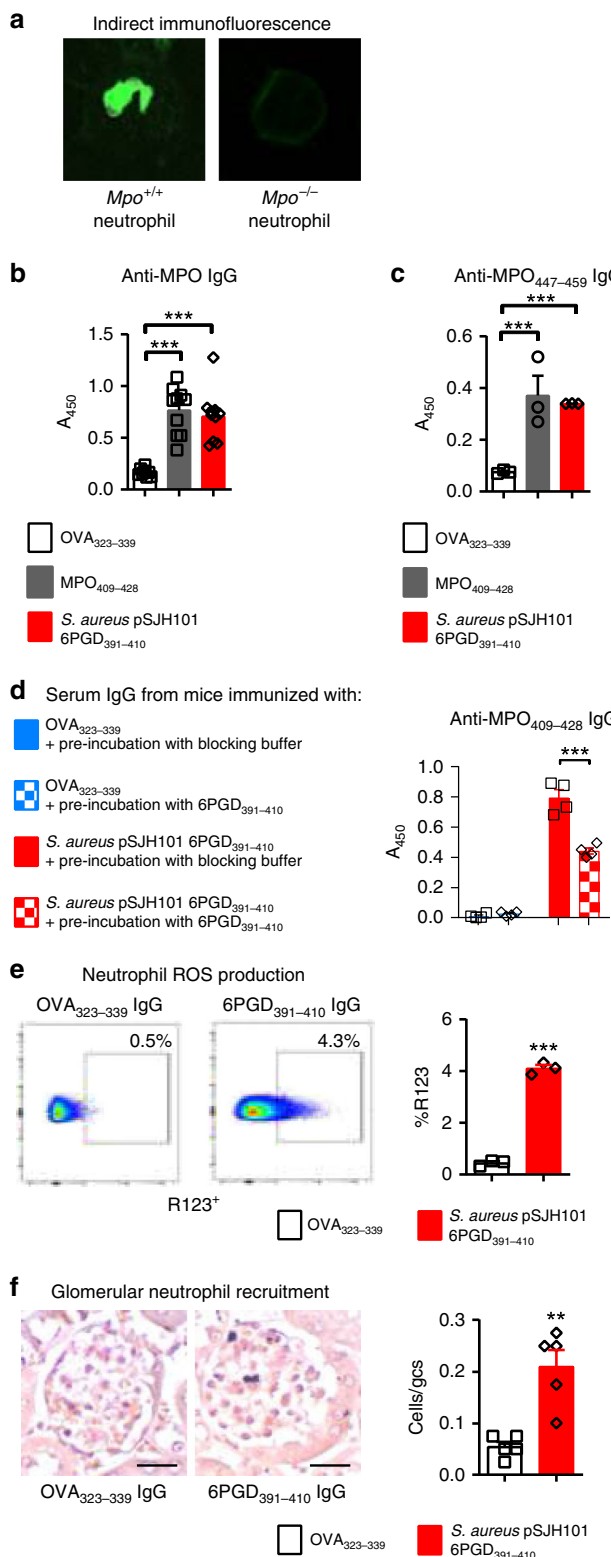

**Fig. 3** MPO-ANCA production in *S. aureus* pSJH101 6PGD$_{391-410}$ immunized mice. **a** Serum IgG from *S. aureus* pSJH101-derived 6PGD$_{391-410}$ immunized C57BL/6 mice (pooled, $n = 8$) binds to neutrophils from C57BL/6 mice, but not those from *Mpo$^{-/-}$* mice in a perinuclear (pANCA) fashion. **b** Anti-MPO ELISA on sera from mice immunized with OVA$_{323-339}$ ($n = 8$), MPO$_{409-428}$ ($n = 10$), or pSJH101 6PGD$_{391-410}$ ($n = 10$, values representative of two independent experiments. **c** Anti-MPO$_{447-459}$ ELISA using pooled serum IgG from pSJH101 6PGD$_{391-410}$-immunized mice, triplicates representative of two independent experiments. **d** *S. aureus* pSJH101 6PGD$_{391-410}$ inhibits autoantibody binding to MPO$_{409-428}$. Serum IgG from *S. aureus* pSJH101 6PGD$_{391-410}$ immunized mice was pre-incubated with *S. aureus* pSJH101 6PGD$_{391-410}$ then used to detect anti-MPO$_{409-428}$ IgG antibodies by ELISA. Values are quadruplicates. **e** Neutrophil reactive oxygen species (ROS) production via rhodamine 123 (R123) induced by pooled serum IgG from pSJH101 6PGD$_{391-410}$ immunized mice. Flow cytometric plots illustrate the data, performed in triplicate. **f** Glomerular neutrophil recruitment after injection of serum IgG from pSJH101 6PGD$_{391-410}$ immunized mice ($n = 5$ each group). Photomicrographs illustrate the data, presented numerically as neutrophils per glomerular cross section (Cells/gcs). Scale bar is 30 μm. Error bars represent mean ± s.e.m. **$P < 0.01$, ***$P < 0.001$ by Kruskal–Wallis test (**b**, **c**), Mann–Whitney $U$-test (**d**, **e**, **f**). Source data are provided as a Source Data file

*S. aureus*. As hypothesized, mice immunized with Variant 3 of 6PGD$_{391-410}$ did not develop disease (Fig. 7), demonstrating the relative specificity of the JH1 pSJH101 6PGD$_{391-410}$ sequence in nephritogenic anti-MPO autoimmunity.

**S. aureus JH1 with pSJH101 immunization leads to nephritis**. To address a specific role for the *S. aureus* pSJH101 plasmid-derived 6PGD$_{391-410}$ sequence in anti-MPO autoimmunity and glomerulonephritis in the context of whole bacteria, we immunized mice with either heat-killed *S. aureus* JH1 strain containing the pSJH101 plasmid or heat-killed JH1 that had been cured of the pSJH101 plasmid (Supplementary Fig. 2a) and induced the same model of glomerulonephritis. Compared to mice immunized with cured heat-killed *S. aureus* JH1, mice immunized with *S. aureus* JH1 containing pSJH101 developed glomerulonephritis with pathological albuminuria, glomerular focal, and segmental necrosis and infiltrates of CD4$^+$ T cells, CD8$^+$ T cells and macrophages (Fig. 8). Mice immunized with *S. aureus* JH1 containing the pSJH101 plasmid also developed MPO-ANCA and MPO-specific secretion of IFN-γ and tumor necrosis factor (TNF) measured in supernatants of cultured splenocytes restimulated with rmMPO (Fig. 8). Therefore, the pSJH101 plasmid containing the cross-reactive *S. aureus* 6PGD sequence is required for anti-MPO cross-reactivity and disease.

**Plasmid and strain independent 6PGD induced anti-MPO immunity**. To determine if it is the specific 6PGD sequence that causes disease independent of other proteins encoded by pSJH101 and independent of the *S. aureus* strain, we cloned 6PGD containing the mimic $_{397}$TDYQEALRDVVA$_{408}$ sequence into the inducible vector pALC2073 (that does not otherwise express 6PGD) to create pALC2073-6PGD. We then transformed a common laboratory *S. aureus* strain (RN4220[39], that contains neither plasmids nor 6PGD $_{397}$TDYQEALRDVVA$_{408}$) with either pALC2073-6PGD or pALC2073 alone. Enhanced expression of 6PGD was confirmed after the induction of anhydrotetracycline (Supplementary Fig. 2b). We immunized mice with heat-killed *S. aureus* RN4220 expressing 6PGD or heat-killed *S. aureus* RN4220 with pALC2073 alone, and disease was again triggered by low-

immunized mice with elevated albuminuria, glomerular segmental necrosis, and inflammatory cell infiltrates (Fig. 7). Furthermore, the pSJH101 6PGD$_{391-410}$ immunized mice developed MPO-ANCA and T-cell reactivity to rmMPO, detected by measuring dermal delayed type hypersensitivity to rmMPO. A further group of mice was immunized with the Variant 3 peptide of 6PGD$_{391-410}$ (Supplementary Table 3), chosen because, of the four variants it was found most frequently in sequenced strains of

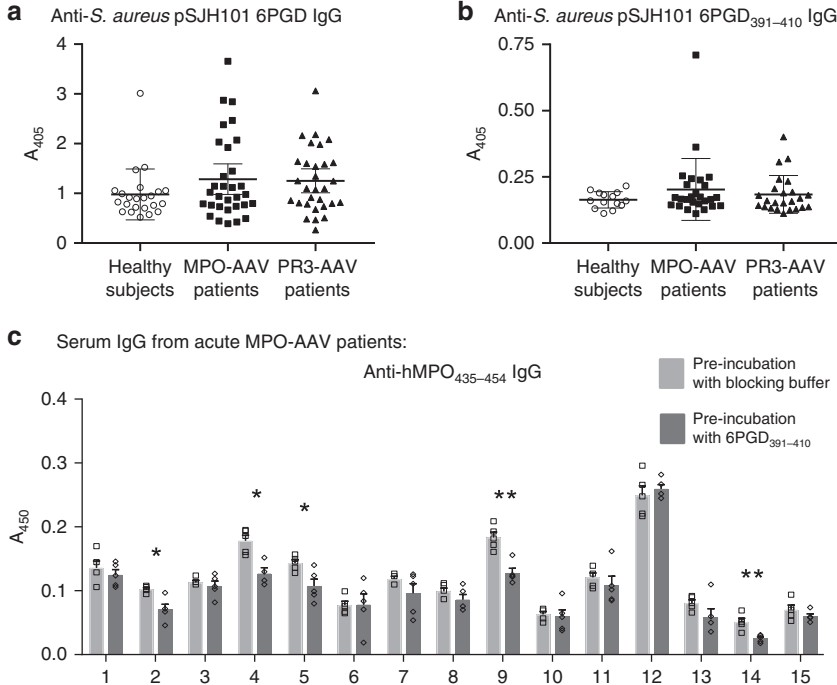

**Fig. 4** Humoral responses to 6PGD and *S. aureus* pSJH101 6PGD$_{391-410}$ in humans. **a** Sera from healthy subjects ($n = 23$), MPO-AAV ($n = 31$) and PR3-AAV patients ($n = 30$) assessed by ELISA for pSJH101-derived recombinant 6PGD. **b** Sera from healthy subjects ($n = 14$), MPO-AAV ($n = 26$) and PR3-AAV patients ($n = 24$) assessed by ELISA to pSJH101 6PGD$_{391-410}$. **c** *S. aureus* pSJH101 6PGD$_{391-410}$ inhibits autoantibody binding to human MPO$_{435-454}$ (the MPO$_{409-428}$ homolog) in acute MPO-AAV. Serum IgG from patients with acute MPO-AAV ($n = 15$) were pre-incubated with *S. aureus* pSJH101 6PGD$_{391-410}$ then used to detect anti-hMPO$_{435-454}$ IgG antibodies by ELISA. Values are quintuplicates. Error bars in **a** and **b** are mean ± s.d., in panel **c** mean ± s.e.m. *$P < 0.05$, **$P < 0.01$, Mann–Whitney *U*-test. Source data are provided as a Source Data file

dose heterologous anti-mouse basement membrane antibodies. Mice immunized with *S. aureus* RN4220 with pALC2073 containing 6PGD developed elevated albuminuria, glomerular segmental necrosis, increases in glomerular CD4$^+$ T cells, CD8$^+$ T cells and macrophages, as well as MPO-specific IgG and MPO-specific splenocyte secretion of IFN-γ and TNF (Fig. 9). Mice immunized with RN4220 with pALC2073 alone were similar to OVA-immunized mice (Fig. 9). Therefore, it is the 6PGD sequence $_{391}$YFKNIVTDYQEALRDVVATG$_{410}$ itself that induces anti-MPO pathogenic autoreactivity, independent of the *S. aureus* strain or plasmid used.

**MHCII promiscuous induction of anti-MPO cross-reactivity.** The dominant MPO T-cell epitope MPO$_{409-428}$, defined in I-A$^b$ expressing C57BL/6 mice is MHCII promiscuous, as MPO$_{409-428}$ also induces autoreactivity in BALB/c mice expressing I-A$^d$/E$^d$ and in humanized HLA transgenic mice expressing HLA-DR15[6]. Here, we show that the core MPO T-cell epitope, MPO$_{415-425}$, previously defined in C57BL/6 mice is the same in both BALB/c and HLA-DR15 mice (Supplementary Fig. 3a, b) and the critical amino acids, defined by alanine substitution, are similar (Supplementary Fig. 3c). To determine if the pSJH101-derived 6PGD$_{391-410}$ induces anti-MPO cross-reactivity in mice expressing different MHCII molecules, we immunized either BALB/c or humanized HLA-DR15 transgenic mice with 6PGD$_{391-410}$ and measured T-cell reactivity, by [$^3$H]-T proliferation assays and ELISPOT for IFN-γ and IL-17A, to 6PGD$_{391-410}$ itself and cross-reactivity to MPO$_{409-428}$ and to rmMPO. We found that both BALB/c and HLA-DR15 transgenic mice developed immunoreactivity to 6PGD$_{391-410}$, and cross-reactivity both to MPO$_{409-428}$ and to rmMPO (Fig. 10a, b), supporting the notion that pSJH101

6PGD$_{391-410}$ sequence can be effectively presented and induce anti-MPO cross-reactivity by a variety of MHCII alleles.

**Discussion**
Although we know that a critical step in the development of autoimmune disease is the activation of pro-inflammatory T cells that react with self-antigens, the steps that precipitate the development and activation of these pathogenic T cells are still unclear. Recently, we have shown that peptide register is a key determinant of the phenotype of the autoreactive T-cell repertoire[40]. While molecular mimicry is often flagged as a potential trigger for the activation of existing autoreactive pro-inflammatory T cells, fewer studies have formally demonstrated microbial-self-peptide cross-reactivity, which is often attributable to the lack of understanding of the self-antigen that precipitates disease[26–28]. The current studies not only identify a mimotope peptide, pSJH101 6PGD$_{391-410}$ that induces anti-MPO T and B-cell autoimmunity, they also highlight both the sensitivity of such mimicry, as very similar sequences to the mimic peptide were unable to induce cross-reactivity. Importantly our studies demonstrate the potential for mimicry to be induced by a plasmid-encoded microbial sequence, identifying a potential new role for bacterial plasmids in the pathogenesis of disease.

We and others have identified a "molecular hotspot" within MPO where an immunodominant T-cell epitope and a disease-associated antibody epitope overlap[6,8,34]. PR3-AAV is classically associated with *S.aureus*[14,15] and reports also implicate *S. aureus* infections in MPO-AAV[19–22]. However, despite the presence of neutrophil-derived MPO at sites of infection in a potentially "dangerous" immunological context, the links between the loss of tolerance to MPO and microbial-derived peptides are unclear. Using a standard and unbiased approach of searching microbial

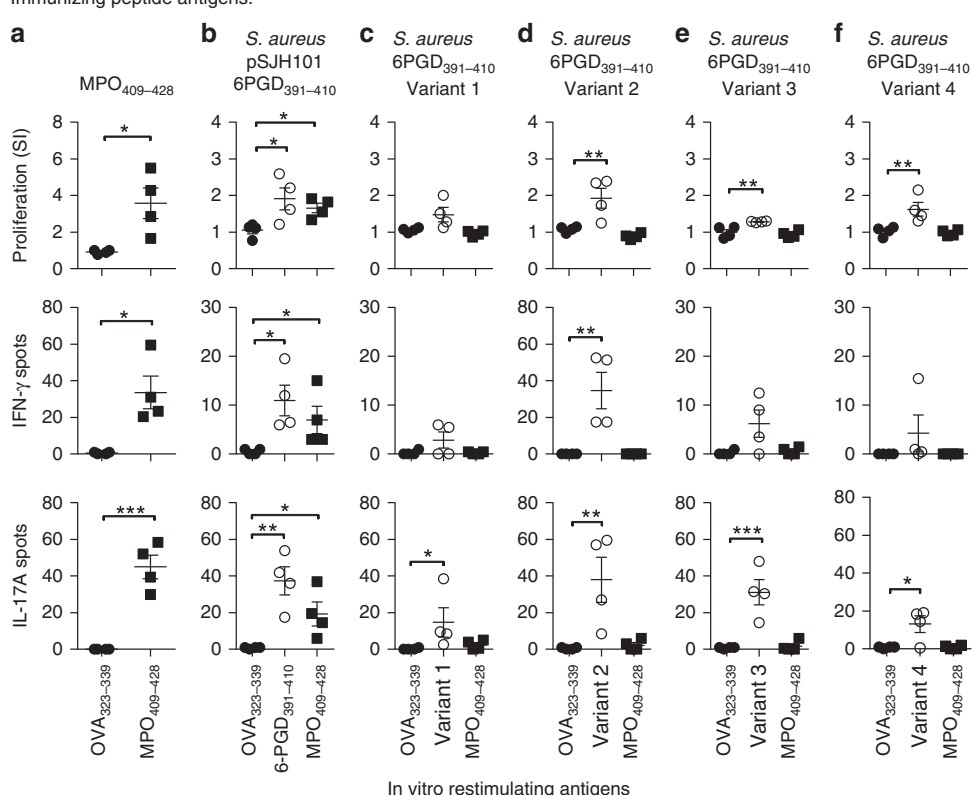

**Fig. 5** Anti-MPO T-cell responses after immunization S. aureus-derived 6PGD$_{391-410}$ sequences. C57BL/6 mice ($n = 4$ each group) were immunized with either **a** MPO$_{409-428}$ (positive control), **b** pSJH101-derived 6PGD$_{391-410}$ (YFKNIVTDYQEALRDVVATG), **c** S. aureus 6PGD$_{391-410}$ Variant 1 (YFKNIVTDYQDALRDVVATG), **d** S. aureus 6PGD$_{391-410}$ Variant 2 (YFKNIVTNYQEALRDVVATG), **e** S. aureus 6PGD$_{391-410}$ Variant 3 (YFKNIVTEYQDALRDVVATG), **f** S. aureus 6PGD$_{391-410}$ Variant 4 (YFKNIVTNYQDALRDVVATG). T-cell recall responses were measured ex vivo to either OVA$_{323-339}$, the immunizing peptide, or MPO$_{409-428}$ using [$^3$H]-thymidine proliferation assays (top row), and ELISPOT IFN-γ (middle row) or IL-17A (bottom row). Each dot represents the response from an individual mouse, error bars represent the mean ± s.e.m. Data are representative of two independent experiments. *$P < 0.05$, **$P < 0.01$, ***$P < 0.001$ by Kruskal–Wallis test. Source data are provided as a Source Data file

proteomes in silico for peptide sequences with the highest sequence similarities to MPO$_{441-451}$ we identified a number of microbial peptides from human pathogens, but experimentally these sequences did not induce anti-MPO cross-reactivity. However, when S. aureus-derived peptides sharing the critical amino acid residues were examined, we identified a plasmid-derived peptide that induces anti-MPO immunoreactivity in the context of several different MHCII molecules and that is immunogenic in humans.

This MPO mimotope, pSJH101 6PGD$_{391-410}$, is overall less homologous than the other non-cross-reactive microbial-derived peptides tested, demonstrating that sequence similarity itself is not necessarily a predictor of molecular mimicry[41]. Instead, specific structural determinants may be more of a contributory factor that leads to cross-reactivity[42]. Our experiments, using similar 6PGD$_{391-410}$ sequences from a range of S. aureus strains demonstrated that even single amino-acid substitutions were sufficient to abrogate anti-MPO cross-reactivity. For example, in Variant 1, a substitution from glutamic acid (E) to the smaller aspartic acid (D), and in Variant 2, a substitution from the negatively charged aspartic acid (D) to the uncharged asparagine (N), prevented the induction of anti-MPO cross-reactivity, highlighting the exquisite sensitivity of TCRs to specific peptide structures.

Using ex vivo restimulation assays, as well as MPO:I-A$^b$ tetramers, we have demonstrated that pSJH101 6PGD$_{391-410}$ can

induce anti-MPO CD4$^+$ T-cell cross-reactivity. Furthermore, in addition to cellular immunity, the 6PGD$_{391-410}$ peptide also induces autoantibodies to whole nmMPO, to the disease-associated linear MPO peptide and to an overlapping linear MPO peptide. The 6PGD$_{391-410}$ mimotope inhibited autoantibody binding to this peptide in mice via a solid phase competitive ELISA. 6PGD$_{391-410}$ also inhibited binding to human MPO$_{435-454}$ (equivalent to mouse MPO$_{409-428}$) in 5/15 (33%) of humans with acute MPO-AAV. Collectively, these data confirm a functional interaction between these overlapping epitopes. Thus, the pSJH101 6PGD$_{391-410}$ peptide cross reacts with an MPO T-cell epitope, but it is also likely to be relevant to these linked B-cell epitopes. While it is possible that antibodies to 6PGD$_{391-410}$ serve as effectors, as for example in the seminal studies of Kaplan and Meyesarian, and others for streptococcal antigens and acute rheumatic fever[43,44], we suggest that this type of direct reactivity at an effector level is less likely in MPO-AAV. Cross-reactivity at a B cell/B-cell receptor level is more likely to be relevant to the promotion of B-cell autoreactivity via binding of 6PGD$_{391-410}$ to the B-cell receptor of potentially autoreactive B cells. This would promote autoreactive anti-MPO B-cell activation by autoreactive CD4$^+$ T cells reacting to the same peptide. In this context, the relative affinities of 6PGD$_{391-410}$ and MPO$_{409-428}$ (in humans MPO$_{435-454}$) to anti-MPO antibodies and whether 100% inhibition occurs, is unlikely to be of critical importance. Furthermore, 6PGD$_{391-410}$ alone is unlikely to have a measurable effect on the

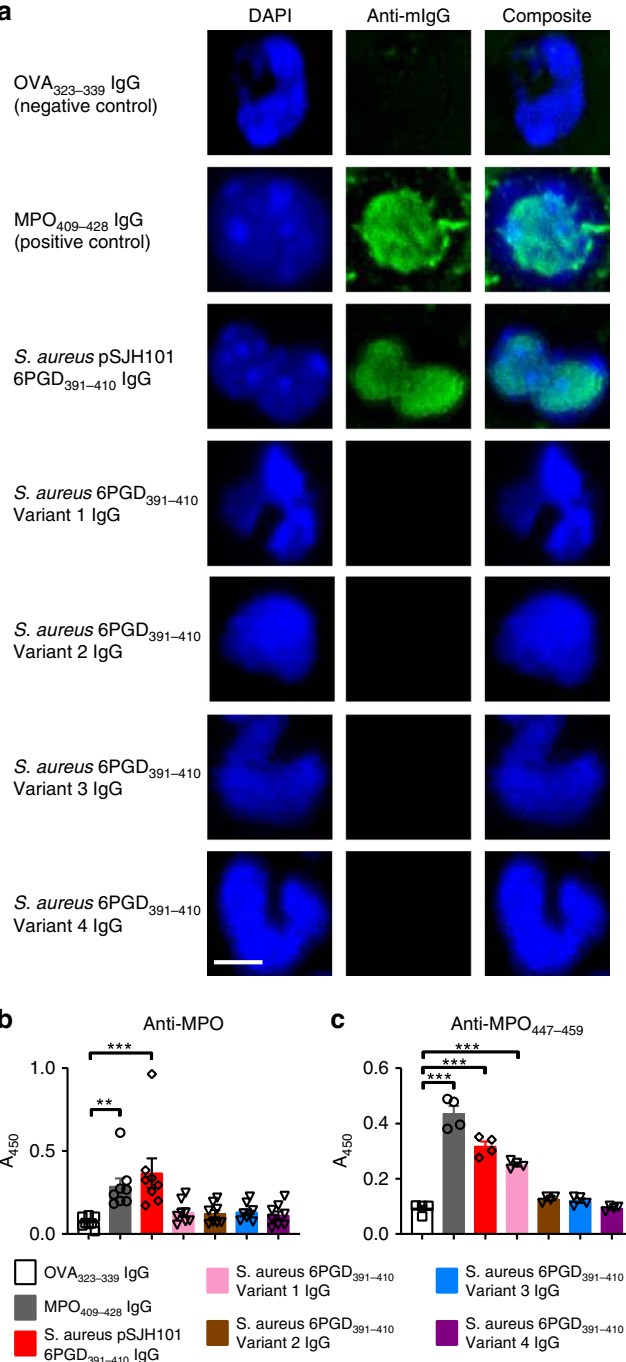

**a**

| | DAPI | Anti-mIgG | Composite |
|---|---|---|---|
| OVA$_{323-339}$ IgG (negative control) | | | |
| MPO$_{409-428}$ IgG (positive control) | | | |
| S. aureus pSJH101 6PGD$_{391-410}$ IgG | | | |
| S. aureus 6PGD$_{391-410}$ Variant 1 IgG | | | |
| S. aureus 6PGD$_{391-410}$ Variant 2 IgG | | | |
| S. aureus 6PGD$_{391-410}$ Variant 3 IgG | | | |
| S. aureus 6PGD$_{391-410}$ Variant 4 IgG | | | |

**b** Anti-MPO

**c** Anti-MPO$_{447-459}$

☐ OVA$_{323-339}$ IgG
■ MPO$_{409-428}$ IgG
■ S. aureus pSJH101 6PGD$_{391-410}$ IgG
■ S. aureus 6PGD$_{391-410}$ Variant 1 IgG
■ S. aureus 6PGD$_{391-410}$ Variant 2 IgG
■ S. aureus 6PGD$_{391-410}$ Variant 3 IgG
■ S. aureus 6PGD$_{391-410}$ Variant 4 IgG

**Fig. 6** MPO-ANCA after immunization with *S. aureus*-derived 6PGD$_{391-410}$ sequences. **a** Perinuclear (pANCA) detection using purified serum IgG, pooled from each group of C57BL/6 mice ($n = 8$ each group) immunized with OVA$_{323-339}$, MPO$_{409-428}$, pSJH101-derived 6PGD$_{391-410}$ (YFKNIVTDYQEALRDVVATG), *S. aureus* 6PGD$_{391-410}$ Variant 1 (YFKNIVTDYQDALRDVVATG), *S. aureus* 6PGD$_{391-410}$ Variant 2 (YFKNIVTNYQEALRDVVATG), *S. aureus* 6PGD$_{391-410}$ Variant 3 (YFKNIVTEYQDALRDVVATG), *S. aureus* 6PGD$_{391-410}$ Variant 4 (YFKNIVTNYQDALRDVVATG) (amino acid substitutions underlined). Nuclei are blue (DAPI), mouse IgG is green (FITC-conjugated anti-mouse IgG antibody). **b** Sera from individual mice ($n = 8$ each group) immunized with pSJH101-derived 6PGD$_{391-410}$ and other *S. aureus*-derived sequences of 6PGD$_{391-410}$ tested by an anti-rmMPO ELISA. **c** Anti- MPO$_{447-459}$ antibody detection by ELISA using purified serum IgG pooled from each group of C57BL/6 mice (**a**). Scale bar is 5 μm (all photomicrographs). Error bars represent the mean ± s.e.m. of triplicates. **$P < 0.01$, ***$P < 0.001$ by Kruskal–Wallis test. Source data are provided as a Source Data file

It is not yet known in humans whether carriage or infection of *S. aureus* strains containing the cross-reactive 6PGD$_{391-410}$ sequence promotes the induction of MPO-AAV or precipitates disease relapse. The conditions for 6PGD$_{391-410}$ recognition to induce anti-MPO T-cell cross-reactivity may include *S. aureus* infection, intermittent colonization or chronic colonization. Furthermore, while nasal swabs are the most common way of screening for *S. aureus*, carriage also occurs on the skin, and in the throat, vagina, anus, and lower gastrointestinal tract[46–48]. It is unlikely that the 6PGD$_{391-410}$ mimotope is the sole factor that determines loss of tolerance to MPO, given the frequency of antibodies to the 6PGD protein and peptide, and the multiple genetic and environmental factors that contribute to the development of MPO-AAV.

Although our data do not conclusively prove a role for 6PGD$_{391-410}$, they suggest that exposure to certain *S. aureus* strains may be a precipitating factor in the loss of tolerance to MPO and the development of MPO-AAV. Our data also demonstrates that plasmids, acting as mobile genetic elements, may transfer a tendency to autoreactivity. The transfer of antibiotic resistance via plasmids is well known. However, the horizontal gene transfer of the cross-reactive 6PGD that we emulated by transforming *S. aureus* RN4220 with pALC2073-6PGD demonstrates that plasmids harboring cross-reactive peptide sequences can induce loss of tolerance. In conclusion, our findings identify pSJH101 6PGD$_{391-410}$ as an MPO cross-reactive mimotope peptide. 6PGD$_{391-410}$ is part of a protein that is immunogenic in humans, can induce loss of tolerance to MPO and experimental anti-MPO glomerulonephritis and MPO-AAV. This sequence is derived from a plasmid found in only some strains of *S. aureus*, implicating plasmid-derived antigens in the loss of tolerance to self-antigens.

## Methods

**Mice**. C57BL/6 and BALB/c mice were obtained from the Monash Animal Research Platform, Clayton, Monash University. *Mpo*$^{-/-}$ mice[49] and HLA-DR15 Tg[50] mice were bred at the Monash Medical Center Animal Facility (MMCAF), Monash Medical Center, Clayton. Mice were housed in the SPF facilities at MMCAF and experiments were conducted in male mice aged 6–10 week. All animal studies were approved by the Monash University Animal Ethics Committee (Committee MMCB) and complied with the Australian code for the care and use of animals for scientific purposes (2013).

**Human samples**. Serum samples from AAV patients and healthy subjects (HS) were obtained from an existing collection of the 'Groningen cohort of AAV', and sera and plasma exchange effluent from Monash patients with acute MPO-AAV were obtained from the Monash Vasculitis Registry and Biobank. Institutional review board (IRB) approval was previously obtained from the Medical Ethics

binding of MPO-ANCA to neutrophils by indirect immunofluorescence, as there are known to be multiple B-cell epitopes in active MPO-AAV[34].

There have been several studies of nasal carriage of *S. aureus* in people with PR3-AAV, due in part to sinonasal disease being common in PR3-AAV[14,15,19]. However, the potential relationship between *S. aureus* and MPO-AAV has been largely unexplored, though colonization with *S. aureus* does occur in patients with this disease[19]. Most *S. aureus* strains known to carry the nephritogenic 6PGD$_{391-410}$ sequence belong to the CC5 clonal complex[19]. In *S. aureus* carriers with established MPO-AAV, 11% of isolates were CC5 (healthy controls 5%, PR3-AAV 15%)[19]. CC5 is a globally distributed clonal complex of *S. aureus* found in both community and hospital settings[35,36,45].

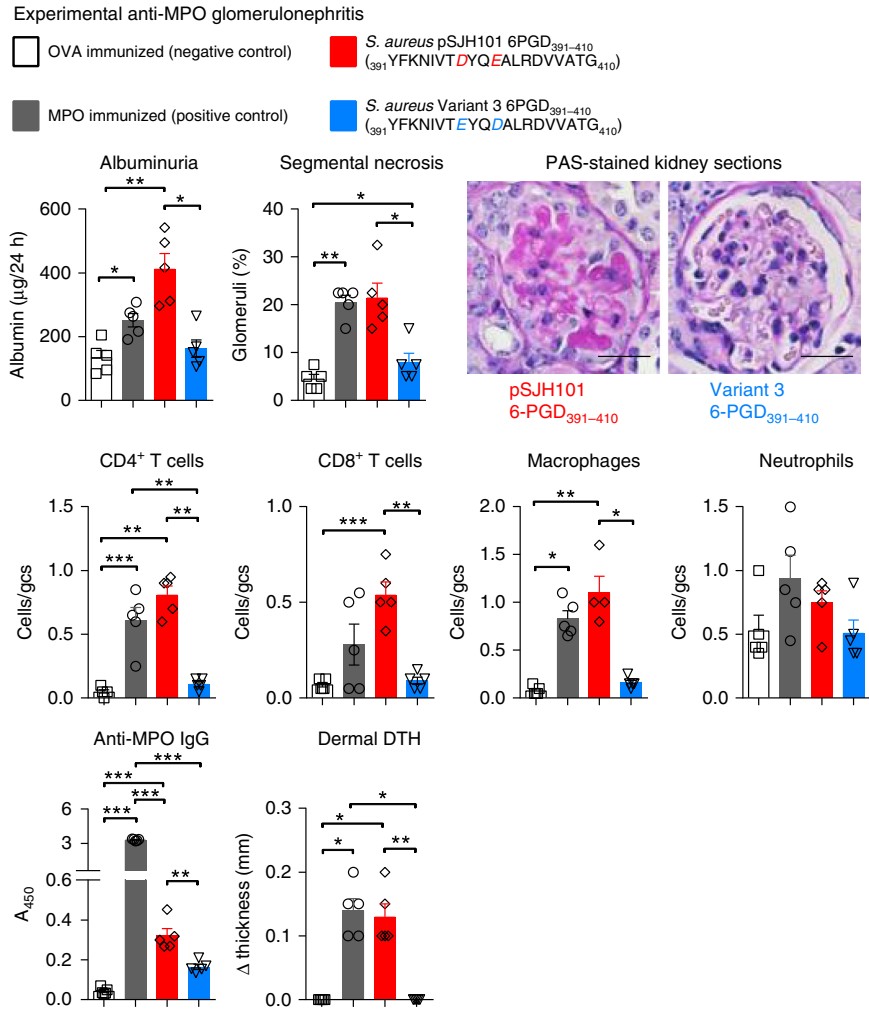

**Fig. 7** Experimental anti-MPO glomerulonephritis in *S. aureus* pSJH101 6PGD$_{391-410}$ immunized mice. C57BL/6 mice ($n = 5$ each group) were immunized with either OVA (negative control), MPO (positive control), *S. aureus* pSJH101 6PGD$_{391-410}$ or the common *S. aureus* 6PGD$_{391-410}$ variant, Variant 3. Low-dose heterologous anti-basement membrane globulin was injected intravenously to induce transient neutrophil recruitment and MPO deposition in glomeruli. Glomerular injury was measured by albuminuria, and by glomerular segmental necrosis on periodic acid-Schiff (PAS) stained kidney sections. Photomicrographs glomeruli from *S. aureus* pSJH101 6PGD$_{391-410}$ or *S. aureus* Variant 3 6PGD$_{391-410}$ immunized mice. Inflammatory cells within glomeruli were enumerated and expressed as cells per glomerular cross section (gcs). Anti-MPO autoreactivity determined by detection of anti-MPO IgG by ELISA and dermal delayed type hypersensitivity (DTH) swelling after recombinant mouse MPO intradermal challenge. Scale bar is 30 μm. Error bars represent the mean ± s.e.m. *$P < 0.05$, **$P < 0.01$, ***$P < 0.001$ by Kruskal–Wallis test. Source data are provided as a Source Data file

Committee of the University Medical Center Groningen and the Monash Health Human Research Ethics Committee, respectively. Written informed consent was obtained from all patients and HS, and all experiments were conducted in accordance with the guidelines of the Declaration of Helsinki. All patients fulfilled the Chapel Hill Consensus Conference definitions for the diagnosis of AAV. All patient samples were confirmed positive for either MPO-ANCA or PR3-ANCA by capture ELISA and indirect immunofluorescence on ethanol fixed neutrophils[51,52].

**Peptides and proteins**. All peptides were synthesized at > 95% purity, confirmed by HPLC (Mimotopes). The residue numbers, in subscript, and peptide sequences, in brackets, of the peptides used are: mouse MPO$_{409-428}$ peptide (PRWNGEK-LYQEARKIVGAMV), *Treponema vincentii* hypothetical protein$_{167-186}$ (LRKQLKRLYKEARKIQKCIP), *Aspergillus fumigatus* HEAT repeat protein$_{825-844}$ (ISALPQRWYQEARKIIFEAA), *Helicobacter pylori* RNA polymerase factor sigma-54$_{163-182}$ (RELDNNELYEEARKIILNLE), *Bacteroides sp.* chloramphenicol O-acetyltransferase$_{104-123}$ (YHEDFETFYQEARKIIDSIP), *S. aureus* pSJH101-derived 6PGD$_{391-410}$ (YFKNIVTDYQEALRDVVATG), *S. aureus* 6PGD$_{391-410}$ Variant 1 (YFKNIVTDYQDALRDVVATG), *S. aureus* 6PGD$_{391-410}$ Variant 2 (YFKNIVT-NYQEALRDVVATG), *S. aureus* 6PGD$_{391-410}$ Variant 3 (YFKNIVTEYQ-DALRDVVATG), *S. aureus* 6PGD$_{391-410}$ Variant 4 (YFKNIVTNY-QDALRDVVATG), *Mus musculus* 6PGD$_{394-413}$ (FFKSAVDNCQDSWRRVIS TGV), and control OVA$_{323-339}$ peptide (ISQAVHAAHAEINEAGR). Sequences of

the shortened MPO peptides are listed in Supplementary Table 3. Immunizations to induce CD4$^+$ T-cell responses were performed with 20-mers containing the core 11 amino acids because MHC class II molecules have open-ended binding pockets and additional amino acids on either side enhances immunoreactivity[6]. MPO was produced using a baculovirus system[53] and OVA was purchased (Sigma-Aldrich). Recombinant *S. aureus* pSJH101 6PGD (Genbank ID: CP000737.1) (GeneArt®, ThermoFisher Scientific) was produced using the Champion pET101 Directional TOPO Expression Kit with BL21 Star (DE3) One Shot chemically competent *E. Coli* (ThermoFisher Scientific). Expression was confirmed by using anti-V5 monoclonal antibodies by western blotting and purified by 6xHis tag elution using nickel resins (Promega).

**Generation of MPO:I-A$^b$ tetramers**. MHCII monomers were produced in High Five insect cells (*Trichoplusia ni* BTI-Tn-5B1-4 cells, Invitrogen) using the baculovirus expression system[40,54,55]. DNA encoding the I-A$^b$ α- and β-chains and the mouse MPO$_{415-428}$ ($_{415}$KLYQEARKIVGAMV$_{428}$), fused to the N-terminus of the β-chain via a flexible linker (SGGGSGSGSAS), were cloned into pFastBac Dual vector and recombinant baculovirus propagated in Sf9 insect cells (*Spodoptera frugiperda*, Invitrogen). The C-termini of the I-A$^b$ α- and β-chains contained enterokinase cleavable Fos and Jun leucine zippers, respectively, to promote correct heterodimeric pairing. The C-terminus of the β-chain also contained a BirA ligase recognition sequence for biotinylation and poly-histidine tag for purification,

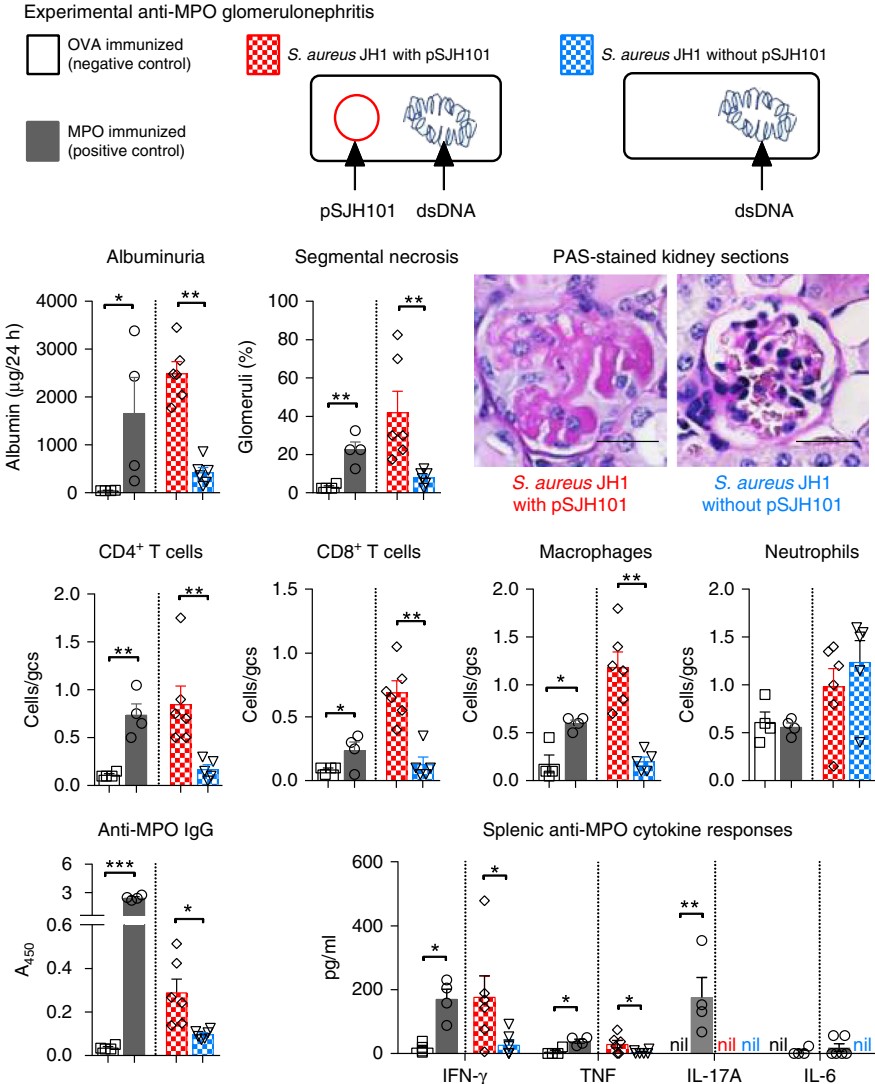

**Fig. 8** Experimental anti-MPO glomerulonephritis in mice injected with *S. aureus* containing pSJH101 6PGD$_{391-410}$. C57BL/6 mice were immunized with either OVA (negative control, $n = 4$), MPO (positive control, $n = 4$), *S. aureus* JH1 with pSJH101 ($n = 6$) or cured *S. aureus* JH1 without pSJH101 ($n = 6$). OVA and MPO were emulsified in Freund's complete adjuvant; *S. aureus* JH1 with or without pSJH101 were emulsified in Titermax. MPO was deposited in glomeruli using heterologous low-dose anti-basement membrane globulin. Glomerular injury was measured by albuminuria, and by glomerular segmental necrosis on periodic acid-Schiff (PAS) stained kidney sections. Photomicrographs depict glomeruli from mice immunized with either *S. aureus* JH1 with pSJH101 or *S. aureus* JH1 without pSJH101. Inflammatory cells within glomeruli were enumerated and expressed as cells per glomerular cross section (gcs). Anti-MPO autoreactivity determined by detection of anti-MPO IgG by ELISA and by measuring inflammatory cytokines, IFN-γ, TNF, IL-17A, and IL-6 in recombinant mouse MPO stimulated splenocyte cultures. Scale bar is 30 μm. Error bars represent the mean ± s.e.m. *$P < 0.05$, **$P < 0.01$, ***$P < 0.001$ by Mann–Whitney *U*-test. Source data are provided as a Source Data file

immediately following the Jun leucine zipper sequence. MPO:I-A$^b$ monomers were purified from baculovirus infected High Five insect cell supernatants through immobilized metal ion affinity (Ni Sepharose 6 Fast-Flow, GE Healthcare), size exclusion (S200 Superdex 16/600, GE Healthcare) and anion exchange (HiTrap Q, HP, GE Healthcare) chromatography. MPO:I-A$^b$ tetramers were assembled by the addition of Streptavidin-PE (BD Biosciences)[54,55].

**Plasmids and *Staphylococcus aureus* strains.** The pSJH101 plasmid was found within a clinical isolate of *S. aureus* JH1 (also known as strain A8090)[56]. To cure the pSJH101 plasmid from *S. aureus* JH1, cells were cultured with 0.004% SDS at 45 °C for 24 h[57]. To confirm the presence or absence of the pSJH101 plasmid containing 6PGD, PCR was performed on cell lysates using primers specific for: *cls2*, forward primer 5′ GCAAGGTACCATGATAGAGTTATTATCCATTGC 3′, reverse primer 5′ GCAAGAGCTCTTAGTGGTGATGGTGATGATGTAAGATAGGTGACAATA ATTGTG 3′; pSJH101, forward primer 5′ CATTGGCGAATCAACAACAC 3′, reverse primer 5′ ACTCCACTTTTGGGGGAACT 3′; and the pSJH101-derived 6PGD, which do not amplify the more common 6PGD (Variant 3) present in the

chromosomal DNA of JH1: forward primer 5′TCATCATCTAACAGCGGAAGT3′ and reverse primer 5′ ACCCCGTAAAATTTTGTTGAT 3′.

The 6PGD sequence (derived from pSJH101) was cloned into the tetracycline inducible pALC2073 plasmid[58]. *S. aureus* RN4220, which contains neither plasmids nor the 6PGD $_{397}$TDYQEALRDVVA$_{408}$ sequence, was transformed by electroporation with either pALC2073 containing 6PGD or pALC2073 without 6PGD[59,60]. To confirm expression of 6PGD we performed PCR on cDNA from cultured *S. aureus* RN4220 containing pALC2073 with 6PGD with or without tetracycline, *S. aureus* RN4220 containing pALC2073 without 6PGD cultured with tetracycline. As a control for specificity, we performed PCR using chromosomal DNA of *S. aureus* RN4220. The primers we used were: forward primer 5′ TCATCA TCTAACAGCGGAAGT 3′ and reverse primer 5′ ACCCCGTAAAATTTTG TTGAT 3′ chromosomal DNA of chromosomal DNA of *S. aureus* RN4220. The primers we used were: forward primer 5′ TCATCATCTAACAGCGGAAGT 3′ and reverse primer 5′ ACCCCGTAAAATTTTGTTGAT 3′. For in silico multi-locus sequence typing (MLST), the software *mlst* was used to identify the sequence types (STs) after scanning the genomes of interest[61], then STs were grouped into CC in which each ST in the CC shares at least six identical alleles of the seven loci with at least one other member of the group[62].

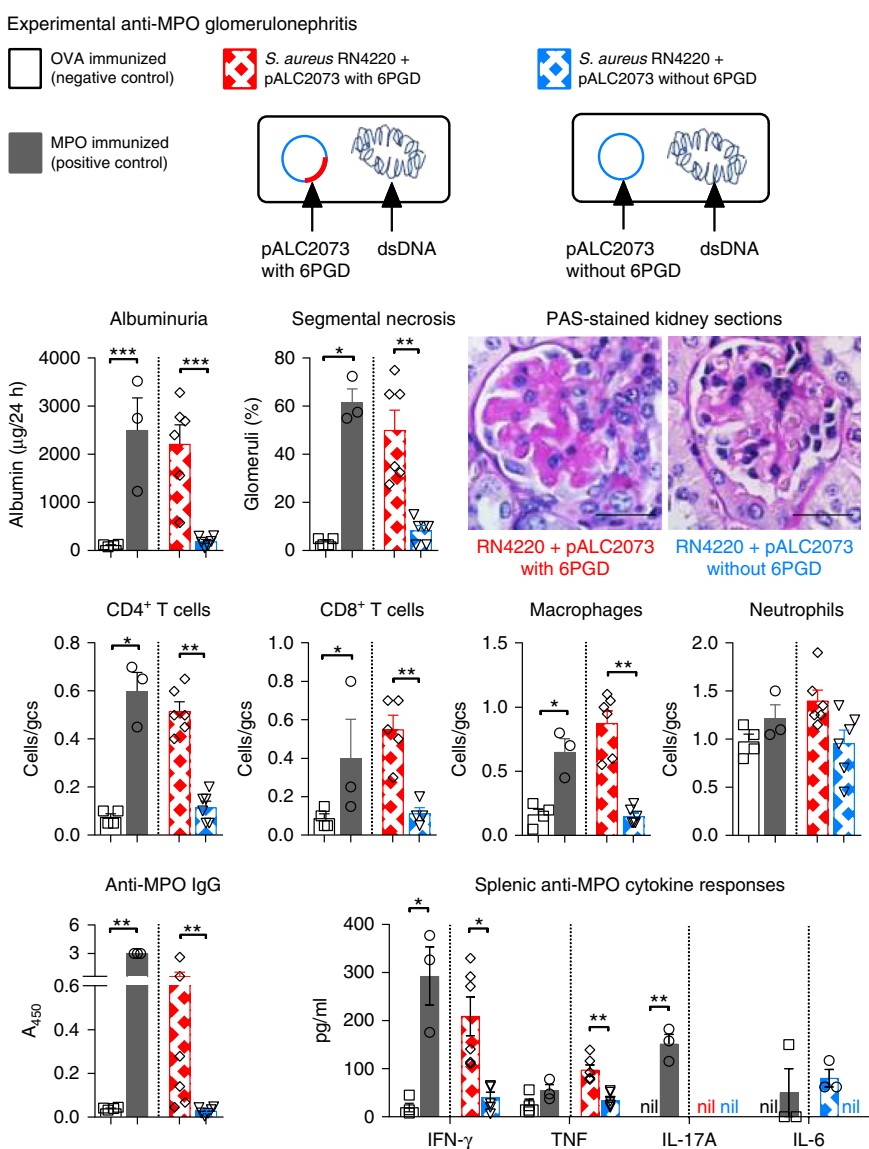

**Fig. 9** Experimental anti-MPO glomerulonephritis in mice injected with *S. aureus* RN4220 containing pALC2073. C57BL/6 mice were immunized with either OVA (negative control, *n* = 4), MPO (positive control, *n* = 3), *S. aureus* RN4220 containing pALC2073 with 6PGD (*n* = 6) or *S. aureus* RN4220 containing pALC2073 alone, without 6PGD (*n* = 6). OVA and MPO were emulsified in Freund's complete adjuvant; *S. aureus* RN4220 containing pALC2073 with or without 6PGD were emulsified in Titermax. MPO was deposited in glomeruli using heterologous low-dose anti-basement membrane globulin. Renal injury was measured by albuminuria, and by glomerular segmental necrosis on periodic acid-Schiff (PAS) stained kidney sections. Photomicrographs depict glomeruli from mice immunized with either *S .aureus* RN4220 containing pALC2073 with 6PGD or RN4220 containing pALC2073 without 6PGD. Inflammatory cells within glomeruli were enumerated and expressed as cells per glomerular cross section (gcs). Anti-MPO autoreactivity determined by detection of anti-MPO IgG by ELISA and by measuring inflammatory cytokines, IFN-γ, TNF, IL-17A, and IL-6 in recombinant mouse MPO stimulated splenocyte cultures. Scale bar is 30 μm. Error bars represent the mean ± s.e.m. *$P < 0.05$, **$P < 0.01$, ***$P < 0.001$ by Mann–Whitney *U*-test. Source data are provided as a Source Data file

**Induction and assessment of T-cell responses**. Mice were immunized with 10 μg of peptide emulsified in Freund's complete adjuvant (FCA) subcutaneously at the base of the tail. Ten days later, draining lymph node cells were isolated and cultured in [³H]-T proliferation assays and/or IFN-γ and IL-17A ELISPOTs. Lymph node cells were cultured in triplicate in supplemented RPMI media (10% vol/vol FCS, 2 mM L-glutamine, 100 U mL⁻¹ penicillin, 0.1 mg mL⁻¹ streptomycin, 50 μM 2-Mercaptoethanol) at $5 \times 10^5$ cells per well in the presence or absence of peptide (10 μg ml⁻¹) or whole protein antigen (10 μg ml⁻¹) in a humidified incubator at 37 °C, 5% CO₂ for 72 h in proliferation assays and 18 h in ELISPOTs. In proliferation assays, [³H]-thymidine was added during the last 16 h of culture and results expressed as a stimulation index. For IFN-γ and IL-17A ELISPOTs (eBioscience, anti-IFN-γ antibodies 551216, 1:250 and 554410, 1:250; anti-IL-17A antibodies 555068, 1:1000 and 555067, 1:1000), spots were developed according to the manufacturer's protocol and results expressed as the mean number of spots minus baseline (media alone). To determine the in vivo expansion of MPO-specific cells,

mice were first immunized with 10 μg of peptide emulsified in FCA subcutaneously at the base of the tail, then, 7 days later, the inguinal, axillary, brachial, cervical, mesenteric, and periaortic lymph nodes and spleen were harvested. Following, tetramer-based magnetic enrichment[63,64], cells were incubated with Live/Dead fixable Near IR Dead Cell Stain (Thermo Scientific) then stained with anti-mouse CD4-Pacific Blue (BioLegend, 100531, 1:400) and "dump" antibodies anti-mouse CD11c (all BioLegend, 117311, 1:100), CD11b (101217, 1:100), F4/80 (123120, 1:100), CD8a (100723, 1:100), B220-Alexa Fluor 488 (103225, 1:100). The MPO:I-A^b tetramer⁺ gate was set based on the CD4⁺ live lymphocyte population (see Supplementary Fig. 4 for gating strategy).

**Induction and assessment of anti-MPO antibody responses**. C57BL/6 mice were immunized with 10 μg of either OVA₃₂₃₋₃₃₉, MPO₄₀₉₋₄₂₈, *S. aureus* pSJH101-derived 6PGD₃₉₁₋₄₁₀, *S. aureus* Variant 1 6PGD₃₉₁₋₄₁₀, *S. aureus* Variant 2

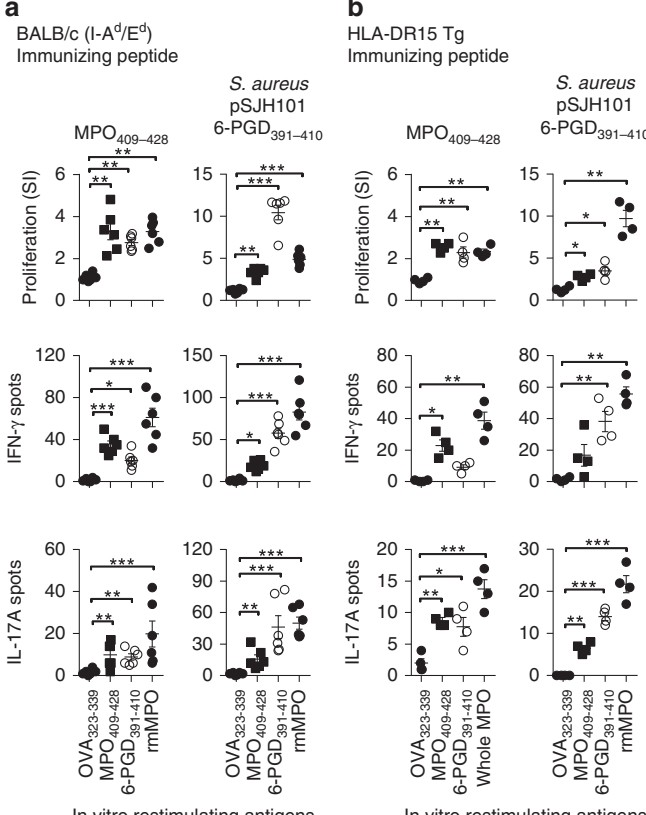

**Fig. 10** Anti-MPO T-cell responses in other *S. aureus* pSJH101 6PGD$_{391–410}$ immunized mouse strains. **a** BALB/c (I-A$^d$/I-E$^d$ expressing, $n = 6$) and **b** HLA-DR15 transgenic (Tg, $n = 4$) mice were immunized with MPO$_{409–428}$ (positive control) and pSJH101 6PGD$_{391–410}$, then T-cell responses measured ex vivo to either OVA$_{323–339}$ (negative control), MPO$_{409–428}$, 6PGD$_{391–410}$, and recombinant mouse MPO using [$^3$H]-thymidine proliferation assays (top row), and ELISOPT for IFN-γ (middle row) or IL-17A (bottom row). Each dot represents the response from an individual mouse, error bars represents the mean ± s.e.m. Data are representative of two independent experiments. *$P < 0.05$, **$P < 0.01$, ***$P < 0.001$ by Kruskal–Wallis test. Source data are provided as a Source Data file

6PGD$_{391–410}$, *S. aureus* Variant 3 6PGD$_{391–410}$ or *S. aureus* Variant 4 6PGD$_{391–410}$; first on day 0 emulsified in (FCA), then boosted on days 7 and 14 emulsified in Freund's incomplete adjuvant (FIA). Serum was collected from mice by cardiac puncture on day 28 and Protein G purified for indirect immunofluorescence on ethanol fixed neutrophils. Thioglycolate induced peritoneal neutrophils were obtained from either *Mpo*$^{+/+}$ or *Mpo*$^{-/-}$ C57BL/6 mice, cytospun onto glass slides then ethanol fixed[6,33]. Pooled serum IgG was incubated with slides for 1 h, then anti-mouse IgG detected using a chicken anti-mouse Alexa Fluor 488 secondary antibody (Molecular Probes, A-21200, 1:200). DAPI was used as a nuclear stain and fluorescence detected by either fluorescence microscopy or confocal microscopy.

**ELISAs for anti-MPO and anti-6PGD antibodies**. Serum was collected from mice by cardiac puncture on day 28 and either used for the detection of anti-MPO IgG antibodies, anti-MPO$_{447–459}$ IgG antibodies by ELISA and inhibition ELISAs for the detection of anti-MPO$_{409–428}$ IgG antibodies. The anti-MPO IgG ELISA was performed on rmMPO coated, 2% casein/PBS blocked 96-well plates. Anti-MPO$_{447–459}$ IgG ELISA was performed on MPO$_{447–459}$ coated, 2% casein/PBS blocked 96-well plates. Serum (diluted 1:50 in PBS) or pooled IgG (100 µg ml$^{-1}$ in PBS) was incubated for 16 h at 4 °C, then anti-mouse IgG detected using a horseradish peroxidase (HRP) conjugated secondary antibody (Amersham, NA-931, 1:2000). For inhibition ELISA, serum IgG (10 µg ml$^{-1}$) was pre-incubated with *S. aureus* pSJH101-derived 6PGD$_{391–410}$ on a 96-well ELISA plate (coating concentration 10 µg ml$^{-1}$), then transferred to an MPO$_{409–428}$ coated (10 µg ml$^{-1}$) 96-well ELISA plate.

Human sera were tested for reactivity to 6PGD (HS $n = 23$, MPO-AAV $n = 31$ and PR3-AAV $n = 30$) and to *S. aureus* pSJH101 6PGD$_{391–410}$ (HS $n = 14$, MPO-AAV $n = 26$) and PR3-AAV patients ($n = 24$) by ELISA. The HS groups were different between assays, and not all samples assayed for whole 6PGD were available for the *S. aureus* pSJH101 6PGD$_{391–410}$ assay. ELISA plates (NUNC Maxisorp, Thermo Fisher Scientific, Breda, The Netherlands) were coated with 100 µl of 5 µg ml$^{-1}$ recombinant *S. aureus* pSJH101 6PGD or 10 µg ml$^{-1}$ *S. aureus* pSJH101 6PGD$_{391–410}$ peptide diluted in 0.1 M carbonate-bicarbonate buffer (pH 9.6) overnight. Plates were washed with PBS pH 7.4 with 0.05% Tween-20 and incubated for 1 h at room temperature (RT) with 200 µl 2% bovine serum albumin (BSA)/PBS per well to prevent non-specific binding. Next, plates were incubated with 100 µl serum samples (1:50 in PBS 1% BSA, 0.05% Tween-20, 2 h at RT). After washing, plates were incubated with alkaline phosphatase goat anti-human IgG (Sigma, St. Louis, USA, A-5403, 1:1000) for one hour at RT and p-nitrophenyl-phosphate disodium (Sigma) was used as a substrate. Absorbance was measured at 405 nm. For inhibition ELISA, IgG purified from sera or plasma exchange effluent (50 µg ml$^{-1}$) was first pre-incubated with *S. aureus* pSJH101-derived 6PGD$_{391–410}$ on a 96-well ELISA plate (coating concentration 10 µg ml$^{-1}$), then transferred to a human MPO$_{435–454}$ coated (10 µg ml$^{-1}$) 96-well ELISA plate.

**Induction of mouse anti-MPO glomerulonephritis**. C57BL/6 mice were immunized subcutaneously at the tail base with either 20 µg of OVA (control antigen), 20 µg of rmMPO, 100 µg of *S. aureus* pSJH101 6PGD$_{391–410}$, 100 µg of *S. aureus* Variant 3 6PGD$_{391–410}$, 10 mg of heat-killed *S. aureus* JH1, 10 mg of cured heat-killed *S. aureus* JH1, 10 mg of heat-killed *S. aureus* RN4220 transformed with pALC2073 with 6PGD or 10 mg of heat-killed *S. aureus* RN4220 transformed with pALC2073 without 6PGD. Proteins and peptides were injected first emulsified in Freund's Complete Adjuvant (FCA) (day 0), then 7 days later emulsified in Freund's Incomplete Adjuvant (FIA) (day 7). *S. aureus* strains were emulsified in Titermax (Sigma-Aldrich) and injected on days 0 and 7. On day 16, MPO was deposited in glomeruli by recruiting neutrophils using a low dose of intravenously injected heterologous anti-mouse basement membrane antibodies[9,38,65]. Experiments ended on day 20. Albuminuria was determined by ELISA (Bethyl Laboratories, E90-134) on urine collected 24 h before the end of experiment. Segmental glomerular necrosis was assessed on formalin fixed, paraffin embedded, 3 µm thick, PAS-stained sections and defined as the accumulation of PAS-positive material with hypocellularity.

CD4$^+$ T cells, CD8$^+$ T cells, macrophages, and neutrophils were detected by immunoperoxidase staining frozen kidney sections. A minimum of 20 consecutively viewed glomeruli were assessed per animal. The primary mAbs used were clones GK1.5 (anti-mouse CD4; American Type Culture Collection, 20 µg ml$^{-1}$), 53–6.7 (anti-CD8a; BioXcell, 10 µg ml$^{-1}$), FA/11 (macrophages, anti-mouse CD68; from GL Koch, MRC Laboratory of Molecular Biology, Cambridge, United Kingdom, 10 µg ml$^{-1}$), and RB6-8C5 (neutrophils, anti-Gr-1, 2.5 µg ml$^{-1}$). MPO-specific delayed type hypersensitivity was measured by intradermal injection of 10 µg of rmMPO, diluted in PBS, into the left plantar footpad. The same volume of PBS was administered into the contralateral footpad. DTH was quantified 24 h later by measurement of the difference in footpad thickness. IFN-γ, TNF, IL-17A, and IL-6 in rmMPO stimulated splenocyte cultures was measured by cytometric bead array (BD Biosciences, 560485).

**Reporting summary**. Further information on research design is available in the Nature Research Reporting Summary linked to this article.

## Data availability
Source data for Figs. 1, 2, 3b–f, 4, 5b–c, 6–10, and Supplementary Figs. 1-3 are presented in the Source Data file. Other data that support the findings of this study are available from the corresponding author upon reasonable request.

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

## Acknowledgements

This work was funded by an Australian National Health and Medical Research Council (NHMRC) Project Grant (1008849 to A.R.K. and S.R.H.), an NHMRC Center for Research Excellence Grant (1079648 to A.R.K.) and an NHMRC European Union collaborative research grant (1115805 to A.R.K.) as part of the EU Horizon 20/20 RELapses prevention in chronic autoimmune disease (RELENT) Consortium (with P.H.). L.F. acknowledges funding from the Wellcome Trust, the Medical Research Council (UK) and the Oak Foundation. A.Y.P. acknowledges funding from an NHMRC Practitioner Fellowship (1117940). J. Rossjohn is supported by an Australian Research Council Laureate Fellowship.

## Author contributions

J.D.O., P.H., A.Y.P., and A.R.K. designed the research and wrote the paper. J.D.O., J.-H.J., P.J.E., L.L.C., M.v.T., K.M.O'S., P.Y.G., and Y.Z. performed and analyzed experiments. K.L.L., H.H.R., and L.F. generated and provided analytical tools. K.T. analyzed data. C.A.S., A.R.K., J. Ryan, and L.R.S. provided samples from healthy humans and people with AAV. S.R.H., L.F., H.H.R. and J. Rossjohn provided intellectual input and technical support.

## Additional information

**Competing interests:** The authors declare no competing interests.

