## [Peer Review File · Nature Communications]

Reviewers' comments:

Reviewer #1 (Remarks to the Author):

In the manuscript by Ooi et al a plasmid encoded peptide from 6PGD present in *S. aureus* is described that induces nephritogenic anti-MPO autoimmunity.

Comments:

1. It is not explained what makes that MPO-AAV patients develop their disease after being exposed to the peptide
2. Whereas there are certainly patients that develop MPO-AAV after a *S. aureus* infection it is not true that nasal carriage is common in MPO-AAV (statement on last sentence of page 6).
3. In the paper by Glasner (ref 15) et al, patients with MPO-AAV might have during follow-up positive nose cultures as is found in healthy controls. Chronic nasal carriage of *s. aureus* is, however, extremely rare in MPO-AAV (see Salmela et al. 2017)..
4. It is not clear what the significance is of Fig 2h. Were positive results obtained in patients and controls that were chronic nasal carrier ? Were positive samples related to intermittent carriage of *s. aureus* ?
5. How do anti-6PGD results relate to positivity to other *S. aureus* proteins ?
6. On page 15, it is mentioned that 11% of *S. aureus* carriers with MPO-AAV are CC5 positive. Unfortunately, data are not given. What about PR3-AAV patients and what about healthy controls ?
7. If only a small minority of *S. aureus* carriers are 6PGD positive, how can be explained that so many patients and controls have anti-6GPD antibodies. Are other tests performed to confirm this reactivity (e.g., immunoblotting) ?
8. Importantly, is there an example of a patient with MPO-AAV that was *S. aureus* carrier with the strain that contained the peptide and developed anti-6GPD antibodies prior to MPO-ANCA ?
9. Finally, can the peptide be demonstrated in renal lesions (either in the murine model or in humans)?

Reviewer #2 (Remarks to the Author):

The article entitled "A staphylococcal plasmid-derived peptide induces anti-myeloperoxidase nephritogenic autoimmunity" by Ooi and Kitching et al reports an extremely interesting and highly novel association between *Staphylococcus aureus* and nephritis mediated through an immune response against the *Staphylococcus* protein 6 phosphogluconate dehydrogenase and myeloperoxidase found in neutrophils. Autoreactivity to myeloperoxidase in neutrophils is induced by a peptide found in a specific *S aureus* strain. The peptide and the strain containing the plasmid with the code for the peptide sequence could produce the anti-neutrophil cytoplasmic antibody associated glomerulonephritis in an established mouse model of glomerulonephritis. The authors have many experiments to prove that the peptide causes the nephritis and induces the autoantibodies and that immunization with the peptide leads to disease and infiltration with neutrophils into the glomeruli. The authors suggest and conclude that a microbial plasmid derived peptide induces experimental anti-myeloperoxidase autoimmunity through molecular mimicry and implicates plasmids as bacterial replicons capable of transferring this sequence in autoimmune disease. The experimental design and results are very clear and the article is well written. The authors will need to address the following comments.

1. The statements about mimicry are suggested by the peptide immunization but are not completely proven in experiments using the peptides. There are several points that need to be addressed about the hypothesis of mimicry. The reason is that since the induction of disease is very narrow, it could suggest that there is some alteration of the basement membrane in the kidney by this strain or by this peptide that leads to the autoimmune response. To address this issue and show that the mimicry is real, the authors must perform competitive inhibition assays using the peptides or the protein in an ELISA or other type of assay such as the immunofluorescence of neutrophils or glomeruli or use several types of assays for the inhibition in order to prove their point about the mimicry. The inhibitions are required since a simple immunization or serum "cross-reactivity" cannot be shown as mimicry without using the antigen in

solution to inhibit the binding of the antibodies to the tissues or inhibiting serum antibody reactivity with the antigen on the ELISA plate for example. The fact that there is no MHC class II restriction makes it all the more important that the inhibitions be performed so the authors can be certain that the cross-reaction is actually real and thus, potentially pathogenic. The use of the peptides which are synthetic should be possible although bacterial proteins can be difficult to prepare and purify, a synthetic peptide of the bacterial protein which works in their studies should be easily studied in competitive inhibitions to show that the cross-reaction really exists.

2. Likewise, the studies of the human sera are not very telling unless the difference between the unaffected controls and the disease serum samples can be understood. This also can be studied using the peptides and should be easy to do with the sera reacting in the ELISA with the antigen and blocking the antibody reaction with antigen with the peptide to show that the cross-reaction occurs in sera. mAbs or T cell clones make the study of molecular mimicry more certain but the competitive inhibitions are good enough with all of the excellent data that the authors show for this antigen and the disease production in the animal model. Perhaps there will be a difference in the healthy control sera and the disease sera in the inhibitions that will be important in understanding the pathogenesis of the nephritis.

3. If the peptides do not inhibit the antibodies binding to antigen or to neutrophils or tissues, then could there be other explanations rather than molecular mimicry for the pathogenesis?

Reviewer #1

We thank the reviewer for their review and comments, which we address in turn below. Modifications to add to and improve the manuscript have been detailed in the responses to the relevant comments.

1. “It is not explained what makes MPO-AAV patients develop their disease after being exposed to the peptide.”

As for most autoimmune diseases, the pathogenesis of AAV is likely to be complex and multifactorial. Current concepts suggest the development of autoimmune disease involves, in the majority of cases, genetic predisposition that sets the scene for the subsequent development of autoimmunity that likely involves one or more significant environmental factors. We have modified the manuscript to state these concepts more explicitly.

The Introduction, Page 4, now reads:

“While it remains unclear how tolerance to neutrophil cytoplasmic antigens MPO and proteinase 3 (PR3) is lost and how disease is triggered [Hutton 2017], like many autoimmune diseases [Goodnow 2007] both genetic and environmental factors are likely to be important [Lyons 2012, Lamprecht 2018].”

The Discussion, Page 17, now reads:

“It is unlikely that the 6PGD₃₉₁₋₄₁₀ mimotope is the sole factor that determines loss of tolerance to MPO, given the frequency of antibodies to the 6PGD protein and peptide, and the multiple genetic and environmental factors that contribute to the development of MPO-AAV.”

2. “Whereas there are certainly patients that develop MPO-AAV after a *S. aureus* infection it is not true that nasal carriage is common in MPO-AAV (statement on last sentence of page 6).”

Thank you for this comment. We have modified statement in the revised manuscript (Introduction page 4 and Results page 7) to read:

Introduction

“Less is known about *S. aureus* colonization of people with MPO-AAV. While chronic nasal carriage is uncommon in those with microscopic polyangiitis and renal limited vasculitis, usually associated with MPO-ANCA [Salmela 2017], nasal colonization does occur [Glasner 2017] and case reports implicate *S. aureus* in the development of this condition [Hellmich 2011, Miranda-Fillooy 2006, Kasami 2009].”

Results

“As *S. aureus* infections can precede the development of MPO-AAV [Hellmich 2011, Miranda-Fillooy 2006, Kasami 2009], they are related to an overlapping form of vasculitis (PR3-AAV) [Stegeman 1994, Pendergraft 2011] and nasal colonization of *S. aureus* has been found in people with MPO-AAV [Glasner 2017] we identified a *S. aureus* derived peptide with sequence homology to human MPO₄₄₁₋₄₅₁ by protein BLAST.”

3. “In the paper by Glasner et al, patients with MPO-AAV might have during follow-up positive nose cultures as is found in healthy controls. Chronic nasal carriage of *S. aureus* is, however, extremely rare in MPO-AAV (see Salmela et al. 2017).”

Thank you for this observation. We note that strictly speaking, the paper of Salmela et al.¹ did not report rates of nasal carriage in PR3-AAV and MPO-AAV. Results with regards to *S. aureus* cultures from the nares were not presented along antigenic lines, but syndromically, i.e. GPA, MPA/RLV. However, this is a useful paper, as we do recognize that as expected, the majority of the GPA patients were PR3-ANCA+, and most MPA/RLV patients were MPO-ANCA+. In addition, in this study, approximately 53% of MPA/RLV patients were *intermittent* carriers of *S.*

aureus. Therefore, although it is likely that from this study, *chronic nasal carriage* is not common in MPO-AAV, we are not able to cite the work of Salmela et al. (or to our knowledge, other studies) to definitively state the frequency of *S. aureus* nasal positivity in MPO-AAV.

However, we do appreciate the overall relevance of the comment, and we have modified the Introduction, citing this useful paper in context (please see our response to Comment #2 above) and to soften the statement regarding nasal carriage and MPO-AAV (Introduction, also please see response to Comment #2 above).

Perhaps in our initial submission we unduly emphasised *chronic nasal carriage* as the likely mechanism by which people who are exposed to *S. aureus* strains with the 6PGD₃₉₁₋₄₁₀ mimotope may go on to develop MPO-AAV. Therefore, we have modified the discussion to clarify that the circumstances by which colonization or infection with *S. aureus* strains expressing the 6PGD₃₉₁₋₄₁₀ mimic might induce cross reactivity are as yet unclear. We detail the options in how 6PGD₃₉₁₋₄₁₀ might be cross-recognized. The Discussion (Page 17) now reads:

“It is not yet known in humans whether carriage or infection of *S. aureus* strains containing the cross-reactive 6PGD₃₉₁₋₄₁₀ sequence promotes the induction of MPO-AAV or precipitates disease relapse. The conditions for 6PGD₃₉₁₋₄₁₀ recognition to induce anti-MPO T cell cross reactivity may include *S. aureus* infection, intermittent colonization or chronic colonization. Furthermore, while nasal swabs are the most common way of screening for *S. aureus*, carriage also occurs on the skin, and in the throat, vagina, anus and lower gastrointestinal tract [Wertheim 2005, Acton 2009, Gagnaire 2107].”

4. “It is not clear what the significance is of Fig 2h. Were positive results obtained in patients and controls that were chronic nasal carriers? Were positive samples related to intermittent carriage of *S. aureus*?”

We have not attempted to link results to nasal carriage. Please see our response to Comment #3 for the clarification of the possible conditions in which 6PGD₃₉₁₋₄₁₀ might act as a mimic. The purpose of Figure 2h (original submission, now Fig 3g in the revised manuscript) is to determine whether the 6PGD protein derived from the *S. aureus* JH1 strain was immunogenic or not (Results text, Page 9). Our finding, that 6PGD is immunogenic, supports the concept that reactivity to 6PGD is relevant to the immune system of humans.

We have performed further experiments in an additional and separate assay that examines reactivity to the pSJH101 6PGD₃₉₁₋₄₁₀ peptide, presented in the revised manuscript as Figure 3h, that now demonstrate immunogenicity of the 6PGD₃₉₁₋₄₁₀ mimotope.

We now state in the revised manuscript (Results page 9):

“Furthermore, sera exhibited reactivity to the pSJH101 JH1 *S. aureus* 6PGD₃₉₁₋₄₁₀ sequence by ELISA (Fig 3h), demonstrating the immunogenicity of this sequence in humans.

5. “How do anti-6PGD results relate to positivity to other *S. aureus* proteins?”

We have not tested reactivity to other *S. aureus* proteins in this work, as the hypothesis was that the potential mimic, JH1 *S. aureus* derived 6PGD was immunogenic. Showing that this protein is immunogenic in humans provides support for the molecular mimicry data.

6. “On page 15, it is mentioned that 11% of *S. aureus* carriers with MPO-AAV are CC5 positive. Unfortunately, data are not given. What about PR3-AAV patients and what about healthy controls?”

These data were presented in the Discussion (and not the Results) as they were derived from Glasner C et al.², (Ref 19 of the revised submission). However, we unintentionally omitted to cite the reference and apologise for this omission. This error has been corrected in the revised

manuscript (Ref 19, revised manuscript). We have added the proportions of isolates from controls and PR3-AAV patients that are CC5 to the Discussion (page 17), that now reads:

Most *S. aureus* strains known to carry the nephritogenic 6PGD₃₉₁₋₄₁₀ sequence belong to the CC5 clonal complex [Glasner 2017]. In *S. aureus* carriers with established MPO-AAV, 11% of isolates were CC5 (healthy controls 5%, PR3-AAV 15%)[Glasner 2017].

7. “If only a small minority of *S. aureus* carriers are 6PGD positive, how can it be explained that so many patients and controls have anti-6GPD antibodies? Are other tests performed to confirm this reactivity (e.g. immunoblotting)?”

A minority of *S. aureus* strains express the pSJH101 6PGD₃₉₇₋₄₀₈ mimotope. We would suggest that lifetime infection rates and or intermittent/chronic carriage of *S. aureus* are high, meaning that it is plausible that antibodies against this 6PGD protein are present in many people. We have clarified the purpose of the data originally presented in Figure 2, now Figure 3g in revised manuscript, in the manuscript and in the response to Comment #5 above. We have performed further ELISAs using the Groningen cohort against the purified linear JH1 6PGD₃₉₁₋₄₁₀ peptide that confirm our initial finding (revised manuscript, Figure 3h).

8. “Importantly, is there an example of a patient with MPO-AAV that was a *S. aureus* carrier with the strain that contained the peptide and developed anti-6GPD antibodies prior to MPO-ANCA?”

We agree that such a patient would indeed provide useful data in support of the involvement of the 6PGD₃₉₇₋₄₀₈ mimotope in MPO-AAV. However, we are not aware of such an example at the present time. To be able to find such an MPO-AAV patient would require the prospective collection of *S. aureus* cultures, with stored sera, in people prior to them developing MPO-AAV. As in Australia the incidence of MPO-AAV is of the order of 5-15 per million cases per year³, it would be either a massive undertaking to collect enough samples prospectively to identify such a patient later, or a very significant stroke of good fortune if a patient in this category could be identified.

9. “Finally, can the peptide be demonstrated in renal lesions (either in the murine model or in humans)?”

We have not looked for the 6PGD₃₉₁₋₄₁₀ peptide within renal lesions. There are three reasons behind this decision

1. Extra-leukocyte MPO is present in moderate amounts in kidneys in both human MPO-AAV⁴ and experimental anti-MPO glomerulonephritis⁵. With the immunological cross reactivity between one or more MPO peptides and 6PGD₃₉₁₋₄₁₀, it would be difficult to generate and then use reagents to detect specifically this sequence in the presence of what would be significant amounts of MPO present outside leukocytes within the kidney.
2. While it is possible that 6PGD₃₉₁₋₄₁₀ might be present in the kidneys of humans with MPO-AAV, this is not required for the effects of the 6PGD₃₉₁₋₄₁₀ on loss of tolerance to MPO, anti-MPO autoimmunity and autoimmune disease. Current evidence (summarised by Hutton et al⁶) supports a model of disease pathogenesis whereby tolerance to MPO is lost in secondary lymphoid organs, perhaps following local infection. MPO is present on neutrophils and MPO-ANCA bind to, and activate neutrophils resulting in the lodging in an active process in target tissues. Here, in addition to inducing injury, neutrophils release the autoantigen (MPO). Experimental evidence shows that MPO-derived peptides can be presented to effector antigen specific T cells^{5,7,8}. Thus 6PGD₃₉₁₋₄₁₀ need not be present in target tissues for it to be important in MPO-AAV.
3. In the murine models of autoimmunity presented in the current manuscript, the biological plausibility of the concept that 6PGD₃₉₁₋₄₁₀ lodged in glomeruli contributes to injury is low. The exposure of mice to 6PGD₃₉₁₋₄₁₀ is via subcutaneous injection of peptide or whole killed bacteria in adjuvant (FCA/FIA for peptide, Titermax for whole bacteria). The effect of

adjuvant is to contain antigen locally, so that little antigen is present intravascularly. Any small amount of antigen that could hypothetically be present in the glomerulus having arrived via the circulation would be difficult to detect (and of doubtful significance) given the amount of antigenically active MPO also present.

We have further explained our well characterized model⁹⁻¹¹ in the Results section of the revised manuscript (Page 11) to now read:

“To determine if the loss of tolerance to MPO induced by *S. aureus* JH1 derived pSJH101 6PGD₃₉₁₋₄₁₀ could result in anti-MPO glomerulonephritis, we used our established model of T cell mediated anti-MPO glomerulonephritis [Ruth 2006, Gan 2010]. In this model, C57BL/6 mice immunized with MPO lose tolerance to MPO but do not develop ANCA of sufficient pathogenicity to induce glomerulonephritis. Therefore, MPO is deposited within the glomerulus via neutrophils transiently recruited by injection of low-dose of heterologous anti-mouse basement membrane globulin. In this context, effector MPO specific T cells recognize MPO peptides and mediate glomerular injury [Ooi 2012, Chang 2017, Ruth 2006]. MPO-immunized mice develop glomerulonephritis with pathological albuminuria and segmental glomerular necrosis.”

Reviewer #2

We thank the reviewer for their constructive review, for stating that the manuscript “**reports an extremely interesting and highly novel association between *Staphylococcus aureus* and nephritis**” and for the positive comments on experimental design, results and clarity.

“1. The statements about mimicry are suggested by the peptide immunization but are not completely proven in experiments using the peptides. There are several points that need to be addressed about the hypothesis of mimicry. The reason is that since the induction of disease is very narrow, it could suggest that there is some alteration of the basement membrane in the kidney by this strain or by this peptide that leads to the autoimmune response.”

We have performed inhibition ELISAs in mouse and in humans, using relevant MPO peptides and the pSJH101 6PGD₃₉₇₋₄₀₈ sequence (see response to the next comment, below).

We do not think that alteration of the glomerular basement membrane leading to an autoimmune response to MPO is a plausible explanation for our findings. The autoantigen (MPO) is not produced by intrinsic glomerular cells and is not present in glomeruli in the absence of inflammation. Currently, the generally accepted model of the pathogenesis of MPO-ANCA associated glomerulonephritis is that initial tolerance to MPO is lost systemically in secondary lymphoid organs the autoantigen is important primarily by being expressed on neutrophils, which are activated by ANCA. These neutrophils then localize to vulnerable tissues such as glomeruli, in an active process. Here, in addition to inducing injury, neutrophils release the autoantigen (MPO). Experimental evidence shows that MPO-derived peptides can be recognised by primed effector antigen specific T cells^{5,7,8}, which effect a further wave of injury.

In addition to a section in the first paragraph of the original Introduction (Page 4), we have further clarified the nature of the model of glomerulonephritis used⁹⁻¹¹ in the Results section of the revised manuscript (Page 11) to now read:

“To determine if the loss of tolerance to MPO induced by *S. aureus* JH1 derived pSJH101 6PGD₃₉₁₋₄₁₀ could result in anti-MPO glomerulonephritis, we used our established model of T cell mediated anti-MPO glomerulonephritis [Ruth 2006, Gan 2010]. In this model, C57BL/6 mice immunized with MPO lose tolerance to MPO but do not develop ANCA of sufficient pathogenicity to induce glomerulonephritis. Therefore, MPO is deposited within the glomerulus via neutrophils transiently recruited by injection of low-dose of heterologous anti-mouse basement membrane globulin. In this context, effector MPO specific T cells recognize MPO peptides and mediate glomerular injury [Ooi 2012, Chang 2017, Ruth 2006]. MPO-immunized mice develop glomerulonephritis with pathological albuminuria and segmental glomerular necrosis.”

[1 Continued] To address this issue and show that the mimicry is real, the authors must perform competitive inhibition assays using the peptides or the protein in an ELISA or other type of assay such as the immunofluorescence of neutrophils or glomeruli or use several types of assays for the inhibition in order to prove their point about the mimicry. The inhibitions are required since a simple immunization or serum “cross-reactivity” cannot be shown as mimicry without using the antigen in solution to inhibit the binding of the antibodies to the tissues or inhibiting serum antibody reactivity with the antigen on the ELISA plate for example. The fact that there is no MHC class II restriction makes it all the more important that the inhibitions be performed so the authors can be certain that the cross-reaction is actually real and thus, potentially pathogenic. The use of the peptides which are synthetic should be possible although bacterial proteins can be difficult to prepare and purify, a synthetic peptide of the bacterial protein which works in their studies should be easily studied in competitive inhibitions to show that the cross-reaction really exists.“

2. “Likewise, the studies of the human sera are not very telling unless the difference between the unaffected controls and the disease serum samples can be understood. This also can be studied using the peptides and should be easy to do with the sera reacting in the ELISA with the antigen and blocking the antibody reaction with antigen with the peptide to show that the cross-reaction occurs in sera. mAbs or T cell clones make the study of molecular mimicry more certain but the competitive inhibitions are good enough with all of the excellent data that the authors show for this antigen and the disease production in the animal model. Perhaps there will be a difference in the healthy control sera and the disease sera in the inhibitions that will be important in understanding the pathogenesis of the nephritis.”

We have performed inhibition ELISAs both in immunised mice and in humans with acute AAV, using relevant MPO peptides and the pSJH101 6PGD₃₉₁₋₄₁₀ sequence. The results are presented on pages 9 and 10 of the revised manuscript, detailed in parts of a new Figure 3, in Supplementary Table 2 and discussed on Page 16. Methods are included on pages 19-20, and 24-25.

Mindful of the possibility that some linear epitopes in this region of MPO can be detected only via purified IgG¹², we used purified IgG from mouse sera and from human samples in these studies.

We show that in pSJH101 6PGD₃₉₁₋₄₁₀ immunized mice, autoantibodies to MPO₄₀₉₋₄₂₈ can be inhibited by pre-incubation with 6PGD₃₉₁₋₄₁₀. Thus, in studies of autoreactivity we have demonstrated T cell responses to MPO and MPO₄₀₉₋₄₂₈ (the latter by expansion of MPO₄₁₅₋₄₂₈:I-A^b positive tetramers as well as by functional reactivity) and reactivity to 6PGD₃₉₁₋₄₁₀. While the primary focus of this study is on T cell epitopes, immunisation with 6PGD₃₉₁₋₄₁₀ induced functional ANCA, anti-MPO antibodies and anti-hMPO₄₄₇₋₄₅₉ antibodies. Anti-mMPO₄₀₉₋₄₂₈ antibodies can be inhibited by the pSJH101 6PGD₃₉₁₋₄₁₀ mimotope. In addition, we have shown the specificity of the pSJH101 6PGD₃₉₁₋₄₁₀ sequence.

The changes in the text of the paper are as follows:

Results, Page 9 now states

“To demonstrate antibody cross-reactivity between 6PGD₃₉₁₋₄₁₀ and MPO₄₀₉₋₄₂₈, we performed an inhibition ELISA. Purified serum IgG from 6PGD₃₉₁₋₄₁₀ immunized mice was pre-incubated with 6PGD₃₉₁₋₄₁₀, then used to detect anti-MPO₄₀₉₋₄₂₈ IgG by ELISA. Serum IgG from *S. aureus* 6PGD₃₉₁₋₄₁₀ immunized mice pre-incubated with *S. aureus* 6PGD₃₉₁₋₄₁₀ had lower antibody titers compared with serum IgG pre-incubated with blocking buffer only (Fig 3d).”

Discussion, Page 16 now reads

“...the 6PGD₃₉₁₋₄₁₀ peptide also induces autoantibodies to whole nmMPO, to the disease associated linear MPO peptide and to an overlapping linear MPO peptide. The 6PGD₃₉₁₋₄₁₀ mimotope inhibited autoantibody binding to this peptide in mice.”

We performed similar inhibition studies in humans with MPO-AAV. As it has been established that anti-MPO epitopes can be restricted to people with acute disease¹², we assembled a new cohort of 15 patients with first presentation acute MPO-AAV from the Monash Vasculitis Clinic (summarized in Supplementary Table 2). In four of these patients, 6PGD₃₉₁₋₄₁₀ inhibited autoantibody reactivity to human MPO₄₃₅₋₄₅₄ (the homologue of mouse MPO₄₀₉₋₄₂₈). The results section (Pages 9-10) now reads:

“To determine whether 6PGD₃₉₁₋₄₁₀ could cross-react with anti-MPO antibodies in acute MPO-AAV, a Monash cohort of 15 patients with acute, active MPO-AAV was assessed (Supplementary Table 2). Purified IgG from these patients was assessed by inhibition ELISA by pre-incubation with 6PGD₃₉₁₋₄₁₀, then antibodies to human MPO₄₃₅₋₄₅₄ (the homologous sequence to mouse MPO₄₀₉₋₄₂₈) were examined by ELISA. Of the 15 patients, four showed significant reduction in anti-human MPO₄₃₅₋₄₅₄ titers after incubation with 6PGD₃₉₁₋₄₁₀ (Fig. 3i).”

The Discussion (Page 16) now reads:

“[after discussion of the mouse inhibition studies]. 6PGD₃₉₁₋₄₁₀ also inhibited binding to human MPO₄₃₅₋₄₅₄ (equivalent to mouse MPO₄₀₉₋₄₂₈) in 4/15 (27%) of humans with acute MPO-AAV. Collectively these data confirm a functional interaction between these overlapping epitopes. While the 6PGD₃₉₁₋₄₁₀ cross reacts with an MPO T cell epitope, it is also likely to be relevant to these linked B cell epitopes.”

The data derived from studies in human sera (the Groningen cohort), that included healthy subjects and people with AAV was presented to demonstrate that the JH1 6PGD protein is immunogenic. We have added to these data by also showing reactivity to the 6PGD₃₉₁₋₄₁₀ peptide. These data have further clarified this aspect of the manuscript (Figure 3h, Results page 9).

“3. If the peptides do not inhibit the antibodies binding to antigen or to neutrophils or tissues, then could there be other explanations rather than molecular mimicry for the pathogenesis?”

It is correct that infections can influence autoimmunity in a number of ways. As we discussed in the Introduction (Page 4) other ways by which infections or infectious proteins/peptides influence autoimmunity, and in particular autoimmune vasculitis, include stimulation of the innate immune system via TLR ligation or other receptors at several stages during loss of tolerance, direct B cell stimulation by bacterial products and superantigen mediated immune activation.

We find that the pJH101 6PGD₃₉₁₋₄₁₀ mimotope did inhibit binding of antibodies to mouse MPO₄₀₈₋₄₂₈ and in some humans with acute MPO-AAV 6PGD₃₉₁₋₄₁₀ inhibited autoantibody binding to human MPO₄₃₅₋₄₅₄. This is useful further corroborative evidence for molecular mimicry. However, we also wish to emphasize that this finding is only part of the substantial evidence that we present in the current studies that effectively demonstrates that molecular mimicry is the only possible explanation for effects of the 6PGD₃₉₁₋₄₁₀ sequence. We designed our experiments so that we could, step by step, isolate the effects of molecular mimicry. The results of our experiments demonstrate that molecular mimicry is the only plausible explanation for our findings. The key findings that show this are:

1. Immunization with the 6PGD₃₉₁₋₄₁₀ mimic in Freund's complete adjuvant not only induced an immune response to itself, but also to MPO₄₀₉₋₄₂₈ and to MPO itself (Figure 2, Figure 3a-c). The responses spanned both cellular and humoral autoimmunity. This immunity is relevant and nephritogenic. Thus, under experimental conditions and in terms of loss of tolerance to MPO the 6PGD₃₉₁₋₄₁₀ peptide was able to act in a similar manner to the nephritogenic MPO peptide.

2. Even minor changes in the 6PGD₃₉₁₋₄₁₀ peptide sequence result in no cross reactivity, despite the same adjuvant being administered, showing that it is the pJH101 6PGD₃₉₁₋₄₁₀ sequence that participates in loss of tolerance to MPO (Figures 4 and 5).
3. The same strain of *S. aureus* (ie the JH1 strain) that was treated so that it did not contain plasmids did not induce nephritogenic autoimmunity to MPO (Figure 7).
4. When we transfected plasmids into a strain of *S. aureus* (RN4220) that is normally plasmid deficient, it was only the strain containing the plasmid with the relevant pJH101 6PGD₃₉₁₋₄₁₀ sequence that induced nephritogenic autoimmunity (Figure 8). In this and the previous experiment, we have dissociated the effects of the mimic sequence from all other effects of *S. aureus*.

Therefore, our first experiments established that cross reactivity occurred, while the subsequent experiments established the specificity of this cross reactivity, at a sequence and plasmid level, in each experiment keeping other experimental conditions the same.

However, we do not imply that molecular mimicry is the only way infections may influence the development of MPO-AAV. Clearly as we state in the Introduction, and have published ourselves in the literature^{6,13,14}, there are a number of ways in which infection might influence this disease. Indeed, the whole killed bacteria experiments (Figures 8 and 9) used an adjuvant that does not itself contain bacterial components, suggesting that pathogen associated molecular patterns within *S. aureus* do play a role in this experimental system, consistent with our previous work on TLR ligation and loss of tolerance to MPO¹³.

References

- 1 Salmela, A., Rasmussen, N., Tervaert, J. W. C., Jayne, D. R. W. & Ekstrand, A. Chronic nasal *Staphylococcus aureus* carriage identifies a subset of newly diagnosed granulomatosis with polyangiitis patients with high relapse rate. *Rheumatology* **56**, 965-72 (2017).
- 2 Glasner, C. *et al.* Genetic loci of *Staphylococcus aureus* associated with anti-neutrophil cytoplasmic autoantibody (ANCA)-associated vasculitides. *Sci Rep* **7**, 12211 (2017).
- 3 Ormerod, A. S. & Cook, M. C. Epidemiology of primary systemic vasculitis in the Australian Capital Territory and south-eastern New South Wales. *Intern. Med. J.* **38**, 816-23 (2008).
- 4 O'Sullivan, K. M. *et al.* Renal participation of myeloperoxidase in antineutrophil cytoplasmic antibody (ANCA)-associated glomerulonephritis. *Kidney Int.* **88**, 1030-1046 (2015).
- 5 Ooi, J. D. *et al.* The immunodominant myeloperoxidase T-cell epitope induces local cell-mediated injury in antimyeloperoxidase glomerulonephritis. *Proc. Natl. Acad. Sci. USA* **109**, E2615-24, (2012).
- 6 Hutton, H. L., Holdsworth, S. R. & Kitching, A. R. ANCA-Associated Vasculitis: Pathogenesis, Models, and Preclinical Testing. *Semin Nephrol* **37**, 418-35 (2017).
- 7 Chang, J. *et al.* CD8+ T Cells Effect Glomerular Injury in Experimental Anti-Myeloperoxidase GN. *J. Am. Soc. Nephrol.* **28**, 47-55 (2017).
- 8 Westhorpe, C. L. V. *et al.* Effector CD4(+) T cells recognize intravascular antigen presented by patrolling monocytes. *Nature Commun.* **9**, 747 (2018).
- 9 Ruth, A. J. *et al.* Anti-neutrophil cytoplasmic antibodies and effector CD4+ cells play nonredundant roles in anti-myeloperoxidase crescentic glomerulonephritis. *J. Am. Soc. Nephrol.* **17**, 1940-49 (2006).
- 10 Gan, P. Y. *et al.* Th17 cells promote autoimmune anti-myeloperoxidase glomerulonephritis. *J. Am. Soc. Nephrol.* **21**, 925-31 (2010).
- 11 Ooi, J. D., Gan, P. Y., Odobasic, D., Holdsworth, S. R. & Kitching, A. R. T cell mediated autoimmune glomerular disease in mice. *Curr. Protoc. Immunol.* **107**, 15 27 11-19 (2014).
- 12 Roth, A. J. *et al.* Epitope specificity determines pathogenicity and detectability in ANCA-associated vasculitis. *J. Clin. Invest.* **123**, 1773-83 (2013).
- 13 Summers, S. A. *et al.* Toll-like receptor 2 induces Th17 myeloperoxidase autoimmunity while Toll-like receptor 9 drives Th1 autoimmunity in murine vasculitis. *Arthritis Rheum.* **63**, 1124-35 (2011).
- 14 Summers, S. A. *et al.* Intrinsic renal cell and leukocyte-derived TLR4 aggravate experimental anti-MPO glomerulonephritis. *Kidney Int.* **78**, 1263-74 (2010).

Reviewers' comments:

Reviewer #1 (Remarks to the Author):

In the revised manuscript by Ooi et al all raised questions are adequately answered. I would like to congratulate with their interesting study.

Reviewer #2 (Remarks to the Author):

In response to the reviewers' suggestions, the authors have made revisions to the study entitled "A Staphylococcal derived peptide induces anti-myeloperoxidase nephritogenic autoimmunity". There are several questions about the response and the revisions which the authors should address.

1. In the methods section, the methods for the ELISA procedure and inhibitions has been placed under Human Samples. This needs to be added to the Human Samples title or a new section should be created for the ELISA and Competitive ELISA.
2. If the authors wanted another reference to a staphylococcal plasmid that contained sequence that encoded gene for a protein that was related to autoimmunity, see reference Kil, K-S et al, Infection and Immunity vol 62: 2440-2449.
3. The study of the mimotope peptide was improved as the authors used the reviewers' suggestions to demonstrate the crossreactivity in the ELISA by using a type of ELISA inhibition. However, absorption was performed against the mimotope peptide on a solid phase ELISA plate to remove the antibody which is the same way the ELISA is performed as in the direct ELISA and does not answer the question about the crossreactivity using soluble antigen. Usually soluble peptide antigen works well to inhibit the binding to another similar antigen on the plate or vice versa. In cases where the affinity for one or the other is too high the inhibition might not work or if it does not recognize the antibody in soluble form or just is some other type for reaction with the solid phase ELISA. This is the point of using soluble antigen. Usually the intact antigen would be placed on the ELISA plate and the soluble whole antigen and its peptides used to inhibit the binding to the whole molecule or other antigen in the direct ELISA. The inhibition and absorption experiments may be done by absorption of constant antibody concentration using the soluble peptide/antigen in varying amounts such as 500 ug/ml to 0.1ug/ml to show a dose response curve. It should work in one or both directions to show that the soluble peptide or soluble whole antigen such as MPO will inhibit the binding to the peptide or whole antigen on the ELISA plate.
4. In addition, the inhibitions could be expanded to study the antibody-neutrophil reactivity which is stained with the antibodies from animal immunizations and disease samples from humans. The antibodies could have been blocked from binding the neutrophil with the peptide and the MPO. If this has been shown before, the authors should so state. This type of inhibition to go along with the ELISA inhibitions is more convincing than the one where they show reduced binding and maybe at best 50% inhibition if you place the serum on the ELISA plate with the solid phase antigen to remove the antibody(Figure 3).
5. If the soluble antigen does not work in the inhibitions then the authors would so state and explain the percent inhibition in figure 3 as shown as approximately 50% or less as it appears. The inhibition on tissues and the neutrophils would be much more convincing with soluble antigen to block antibody binding to tissues and in the ELISA.
6. On the figure 3i of the human sera inhibitions with the mimotope peptide, there were 5 sera that could be inhibited, 4 significantly. The fifth sera should be mentioned that it was reduced but not significant.
7. Only 5 of the 15 human sera from disease were inhibited by the peptide. What do the authors think this means? Are there other epitopes? If the soluble antigen inhibitions are successful, then the other sera may be inhibited by soluble antigen and different from those that are inhibited by absorption with antigen bound to the plate.
8. Although the work is highly novel and interesting, the cross reactivity should be given more thought and effort to make it more convincing if possible.

Reviewer #2

We thank the reviewer for their further review and comments.

There are several questions about the response and the revisions which the authors should address.

1. “In the methods section, the methods for the ELISA procedure and inhibitions has been placed under Human Samples. This needs to be added to the Human Samples title or a new section should be created for the ELISA and Competitive ELISA.”

Thank you for this suggestion. We have revised the Methods section by adding a new section where we add a description of the methods for the ELISA and competitive ELISAs (pages 24-25 in the Methods section of the revised manuscript), reading as follows:

“ELISAs for anti-MPO and anti-6PGD antibodies

Serum was collected from mice by cardiac puncture on day 28 and either used for the detection of anti-MPO IgG antibodies, anti-MPO₄₄₇₋₄₅₉ IgG antibodies by ELISA and inhibition ELISAs for the detection of anti-MPO₄₀₉₋₄₂₈ IgG antibodies. The anti-MPO IgG ELISA was performed on rmMPO coated, 2% casein/PBS blocked 96-well plates. Anti-MPO₄₄₇₋₄₅₉ IgG ELISA was performed on MPO₄₄₇₋₄₅₉ coated, 2% casein/PBS blocked 96-well plates. Serum (diluted 1:50 in PBS) or pooled IgG (100 µg ml⁻¹ in PBS) was incubated for 16 h at 4°C, then anti-mouse IgG detected using a horseradish peroxidase (HRP) conjugated secondary antibody (Amersham). For inhibition ELISA, serum IgG (10 µg ml⁻¹) was pre-incubated with *S. aureus* pSJH101 derived 6PGD₃₉₁₋₄₁₀ on a 96-well ELISA plate (coating concentration 10 µg/ml), then transferred to an MPO₄₀₉₋₄₂₈ coated (10 µg ml⁻¹) 96-well ELISA plate.

Human sera were tested for reactivity to 6PGD (HS *n* = 23, MPO-AAV *n* = 31 and PR3-AAV *n* = 30) and to *S. aureus* pSJH101 6PGD₃₉₁₋₄₁₀ (HS *n* = 14, MPO-AAV *n* = 26) and PR3-AAV patients (*n* = 24) by ELISA. The HS groups were different between assays, and not all samples assayed for whole 6PGD were available for the *S. aureus* pSJH101 6PGD₃₉₁₋₄₁₀ assay. ELISA plates (NUNC Maxisorp, Thermo Fisher Scientific, Breda, The Netherlands) were coated with 100 µl of 5 µg ml⁻¹ recombinant *S. aureus* pSJH101 6PGD or 10 µg/ml *S. aureus* pSJH101 6PGD₃₉₁₋₄₁₀ peptide diluted in 0.1 M carbonate-bicarbonate buffer (pH 9.6) overnight. Plates were washed with PBS pH 7.4 with 0.05% Tween-20 and incubated for 1 h at room temperature (RT) with 200 µl 2% bovine serum albumin (BSA)/PBS per well to prevent non-specific binding. Next, plates were incubated with 100 µl serum samples (1:50 in PBS 1% BSA, 0.05% Tween-20, 2 h at RT). After washing, plates were incubated with alkaline phosphatase anti-human IgG (Sigma, St. Louis, USA) for one hour at RT and p-nitrophenyl-phosphate disodium (Sigma) was used as a substrate. Absorbance was measured at 405 nm. For inhibition ELISA, IgG purified from sera or plasma exchange effluent (50 µg ml⁻¹) was first pre-incubated with *S. aureus* pSJH101 derived 6PGD₃₉₁₋₄₁₀ on a 96-well ELISA plate (coating concentration 10 µg ml⁻¹), then transferred to a human MPO₄₃₅₋₄₅₄ coated (10 µg ml⁻¹) 96-well ELISA plate.”

2. “If the authors wanted another reference to a staphylococcal plasmid that contained sequence that encoded gene for a protein that was related to autoimmunity, see reference Kil, K-S et al, Infection and Immunity vol 62: 2440-2449.”

Thank you for this suggestion. We understand this paper examines an antigen found in pathogenic streptococci, and have cited this reference in the revised manuscript (new reference 44) on Page 16 of the discussion as follows:

“While it is possible that antibodies to 6PGD₃₉₁₋₄₁₀ serve as effectors, as for example by the seminal studies of Kaplan and Meyesian, and others for streptococcal antigens and acute

rheumatic fever [Kaplan 1962, Kil 1994], we suggest that this type of direct reactivity at an effector level is less likely in MPO-AAV.”

3. “The study of the mimotope peptide was improved as the authors used the reviewers’ suggestions to demonstrate the crossreactivity in the ELISA by using a type of ELISA inhibition. However, absorption was performed against the mimotope peptide on a solid phase ELISA plate to remove the antibody which is the same way the ELISA is performed as in the direct ELISA and does not answer the question about the crossreactivity using soluble antigen. Usually soluble peptide antigen works well to inhibit the binding to another similar antigen on the plate or vice versa. In cases where the affinity for one or the other is too high the inhibition might not work or if it does not recognize the antibody in soluble form or just is some other type for reaction with the solid phase ELISA. This is the point of using soluble antigen. Usually the intact antigen would be placed on the ELISA plate and the soluble whole antigen and its peptides used to inhibit the binding to the whole molecule or other antigen in the direct ELISA. The inhibition and absorption experiments may be done by absorption of constant antibody concentration using the soluble peptide/antigen in varying amounts such as 500 ug/ml to 0.1ug/ml to show a dose response curve. It should work in one or both directions to show that the soluble peptide or soluble whole antigen such as MPO will inhibit the binding to the peptide or whole antigen on the ELISA plate.”

Thank you for these further comments. The question that was raised in the previous review related to cross reactivity between the mimic 6PGD₃₉₁₋₄₁₀ and MPO₄₀₉₋₄₂₈, with a helpful specific suggestion that we assess cross reactivity in antibodies in mouse and human samples. These types of assays effectively serve as surrogates to test whether the 6PGD sequence (as peptide or larger fragment of protein) could bind to the B cell receptor and lead to simulation of B cells that had some specificity for MPO.

While the primary focus of our studies were T cell responses and molecular mimicry, as a relevant B cell epitope overlaps the immunodominant T cell epitope^{1,2} the 6PGD₃₉₁₋₄₁₀ mimic peptide is likely to have relevance to B cell responses. However, importantly, our manuscript does not intend to claim that antibodies directed against the 6PGD sequence would cross react with MPO-ANCA and themselves act as effectors, as may be the case in some situations such as the classical studies of Kaplan and Mayesarian (ref 43 of revised paper) in rheumatic fever. We have clarified this by rewriting and adding to the manuscript.

The Discussion, Pages 16-17, now reads

“The 6PGD₃₉₁₋₄₁₀ mimotope inhibited autoantibody binding to this peptide in mice via a solid phase competitive ELISA.”

“Thus the pSJH101 6PGD₃₉₁₋₄₁₀ peptide cross reacts with an MPO T cell epitope, but it is also likely to be relevant to these linked B cell epitopes. While it is possible that antibodies to 6PGD₃₉₁₋₄₁₀ serve as effectors, as for example by the seminal studies of Kaplan and Meyesarian, and others for streptococcal antigens and acute rheumatic fever [Kaplan 1962, Kil 1994], we suggest that this type of direct reactivity at an effector level is less likely in MPO-AAV. Cross reactivity at a B cell/B cell receptor level is more likely to be relevant to the promotion of B cell autoreactivity via binding of 6PGD₃₉₁₋₄₁₀ to the B cell receptor of potentially autoreactive B cells. This would promote autoreactive anti-MPO B cell activation by autoreactive CD4⁺ T cells reacting to the same peptide. In this context, the relative affinities of 6PGD₃₉₁₋₄₁₀ and MPO₄₀₉₋₄₂₈ (in humans MPO₄₃₅₋₄₅₄) to anti-MPO antibodies and whether 100% inhibition occurs, is unlikely to be of critical importance. Furthermore, 6PGD₃₉₁₋₄₁₀ alone is unlikely to have a measurable effect on the binding of MPO-ANCA to neutrophils by indirect immunofluorescence, as there are known to be multiple B cell epitopes in active MPO-AAV [Roth 2013].”

In our revision, we performed inhibition ELISAs using the two relevant sequences and showed in the mouse system, where the nature and timing of immunogen exposure can be standardized, that anti-6PGD₃₉₁₋₄₁₀ antibodies did cross react with anti-MPO₄₀₉₋₄₂₈ antibodies. In humans with MPO-AAV, where clearly the duration of autoimmunity is variable and the “epitope profile” is more complex, we showed that antibodies from a proportion of patients showed cross reactivity.

We accept that there are a number of ways by which cross reactivity may be assessed, and acknowledge the reviewer’s preference for a fluid phase competitive ELISA that would assess the relative affinities of antibodies against the 6PGD mimic peptide and MPO₄₀₉₋₄₂₈. However, detailed assessment and analyses of these subtleties, while interesting, is not required to demonstrate molecular mimicry. We contend that we have shown this convincingly in our current manuscript.

We would like to emphasise that we have examined mimicry in anti-MPO autoimmunity via multiple approaches in an integrated manner. The synergism of these approaches, concordance of our data and our revision collectively strongly support cross reactivity as the reason why the JH1 pJH101 6PGD₃₉₁₋₄₁₀ sequence is a mimic. In addition to our revisions that show cross reactivity, we have effectively excluded other aspects of infection mediated loss of tolerance. Unlike at least some studies of molecular mimicry, our experiments have examined the question from multiple angles, with the results demonstrating that molecular mimicry is the only plausible explanation for our findings. In addition to our revisions showing humoral cross reactivity, our experiments have effectively isolated molecular mimicry as the explanation for our findings:

1. Immunization with the 6PGD₃₉₁₋₄₁₀ mimic in adjuvant not only induced a cellular and humoral response not only to itself, but also to MPO₄₀₉₋₄₂₈ and to MPO itself (Figure 2, Figure 3a-c). Further experiments showed this immunity is functionally relevant.
2. Minor changes in the 6PGD₃₉₁₋₄₁₀ peptide sequence result in no cross reactivity, despite the same adjuvant being administered, showing that it is the pJH101 6PGD₃₉₁₋₄₁₀ sequence that induces loss of tolerance to MPO (Figures 4 and 5).
3. Further studies dissociated the effects of the mimic sequence from all other effects of *S. aureus*.
 - a. The JH1 strain of *S. aureus* (ie the strain that was treated so that it did not contain plasmids) did not induce nephritogenic autoimmunity to MPO, despite adjuvant and PAMP stimulation by the *S. aureus* itself (Figure 7).
 - b. When plasmids were transfected into a plasmid deficient strain of *S. aureus* (RN4220), it was only the strain containing the plasmid with the relevant pJH101 6PGD₃₉₁₋₄₁₀ sequence that induced nephritogenic autoimmunity (Figure 8).

4. “In addition, the inhibitions could be expanded to study the antibody-neutrophil reactivity which is stained with the antibodies from animal immunizations and disease samples from humans. The antibodies could have been blocked from binding the neutrophil with the peptide and the MPO. If this has been shown before, the authors should so state. This type of inhibition to go along with the ELISA inhibitions is more convincing than the one where they show reduced binding and maybe at best 50% inhibition if you place the serum on the ELISA plate with the solid phase antigen to remove the antibody (Figure 3).”

These might be interesting experiments to try to perform, if there was a single B cell epitope in this disease that was responsible for the indirect immunofluorescent staining of normal neutrophils. However, autoantibodies against multiple linear and confirmation epitopes are responsible for the P-ANCA pattern seen on indirect immunofluorescence. Evidence for multiple epitopes comes from a number of reports and demonstrated in the paper of Roth AJ et al (J Clin Invest 2013, cited as ref 34 in the revised manuscript) (below).

Figure 1A from Roth et al J Clin Invest 2013.

[redacted]

We have added to the Discussion as follows (Page 17)

“Furthermore, 6PGD₃₉₁₋₄₁₀ alone is unlikely to have a measurable effect on the binding of MPO-ANCA to neutrophils by indirect immunofluorescence, as there are known to be multiple B cell epitopes in active MPO-AAV [Roth et al 2013].”

5. “If the soluble antigen does not work in the inhibitions then the authors would so state and explain the percent inhibition in Figure 3 as shown as approximately 50% or less as it appears. The inhibition on tissues and the neutrophils would be much more convincing with soluble antigen to block antibody binding to tissues and in the ELISA.”

We have commented on this in the revised manuscript, and have explained that it is not necessary, when one is considering the initiation of and development of B cell responses, for antibody inhibition studies as a reasonable surrogate, to demonstrate 100% inhibition. With

regards to antibody binding to tissues, ANCA-associated vasculitis is an interesting disease in that antibodies are not commonly found in tissues, such that the histopathological description of ANCA-associated glomerulonephritis is as a “pauci-immune” disease. ANCA bind to neutrophils, giving the characteristic patterns on indirect immunofluorescence. However, given the number of epitopes in humans with this disease, it is not likely that inhibiting one epitope will result in a decreased signal.

We have explained this in the revised manuscript (see above, responses to #5) and added a clarification as to how ANCA induce injury to the Introduction (Page 4) that now reads:

“In MPO-AAV, tissue injury is induced not by autoantibodies binding to target tissues such as the kidney, but by anti-MPO autoantibodies (MPO-ANCA) that bind to and activate neutrophils causing glomerular neutrophil recruitment, degranulation and NETosis [Kessenbrock et al 2009, O’Sullivan et al 2015, Huugen et al 2005].”

6. On the figure 3i of the human sera inhibitions with the mimetope peptide, there were 5 sera that could be inhibited, 4 significantly. The fifth sera should be mentioned that it was reduced but not significant.

Thank you for prompting us to examine the results from this patient. The reduction in Patient 4 is significant by Mann Whitney U test. We have corrected Figure 3i, and the text of the Results and Discussion.

The text of the Results (Page 10), now reads

“Of the 15 patients, five showed a significant reduction in anti-human MPO₄₃₅₋₄₅₄ titers after incubation with 6PGD₃₉₁₋₄₁₀ (Fig. 3i).”

The Discussion (Page 16) now reads

“6PGD₃₉₁₋₄₁₀ also inhibited binding to human MPO₄₃₅₋₄₅₄ (equivalent to mouse MPO₄₀₉₋₄₂₈) in 5/15 (33%) of humans with acute MPO-AAV.”

7. Only 5 of the 15 human sera from disease were inhibited by the peptide. What do the authors think this means? Are there other epitopes? If the soluble antigen inhibitions are successful, then the other sera may be inhibited by soluble antigen and different from those that are inhibited by absorption with antigen bound to the plate.

There are other epitopes, and the subtleties/fine detail of the B cell epitopes differ between patients (see for example Roth et al J Clin Invest 2013, Fig 1 above and cited as ref 34 in the revised manuscript). We have noted the existence of multiple B cell epitopes in the Discussion (Page 17), see response to #5.

8. Although the work is highly novel and interesting, the cross reactivity should be given more thought and effort to make it more convincing if possible.

We appreciate the comment. We have given considerable further thought to cross reactivity and have added to and modified the paper to clarify and enhance the quality of the manuscript.

References

- 1 Ooi, J. D. *et al.* The immunodominant myeloperoxidase T-cell epitope induces local cell-mediated injury in antimyeloperoxidase glomerulonephritis. *Proc Natl Acad Sci U S A* **109**, E2615-2624, (2012).
- 2 Roth, A. J. *et al.* Epitope specificity determines pathogenicity and detectability in ANCA-associated vasculitis. *J Clin Invest* **123**, 1773-1783 (2013).